# The cytidine deaminase APOBEC3A regulates nucleolar function to promote cell growth and ribosome biogenesis

**Mason A. McCool**[1], **Carson J. Bryant**[1], **Laura Abriola**[2], **Yulia V. Surovtseva**[2], **Susan J. Baserga**[1,3,4]*

**1** Department of Molecular Biophysics & Biochemistry, Yale University School of Medicine, New Haven, Connecticut, United States of America, **2** Yale Center for Molecular Discovery, Yale University, West Haven, Connecticut, United States of America, **3** Department of Genetics, Yale University School of Medicine, New Haven, Connecticut, United States of America, **4** Department of Therapeutic Radiology, Yale University School of Medicine, New Haven, Connecticut, United States of America

* susan.baserga@yale.edu

## Abstract

Cancer initiates as a consequence of genomic mutations and its subsequent progression relies in part on increased production of ribosomes to maintain high levels of protein synthesis for unchecked cell growth. Recently, cytidine deaminases have been uncovered as sources of mutagenesis in cancer. In an attempt to form a connection between these 2 cancer driving processes, we interrogated the cytidine deaminase family of proteins for potential roles in human ribosome biogenesis. We identified and validated APOBEC3A and APOBEC4 as novel ribosome biogenesis factors through our laboratory's established screening platform for the discovery of regulators of nucleolar function in MCF10A cells. Through siRNA depletion experiments, we highlight APOBEC3A's requirement in making ribosomes and specific role within the processing and maturation steps that form the large subunit 5.8S and 28S ribosomal (r)RNAs. We demonstrate that a subset of APOBEC3A resides within the nucleolus and associates with critical ribosome biogenesis factors. Mechanistic insight was revealed by transient overexpression of both wild-type and a catalytically dead mutated APOBEC3A, which both increase cell growth and protein synthesis. Through an innovative nuclear RNA sequencing methodology, we identify only modest predicted APOBEC3A C-to-U target sites on the pre-rRNA and pre-mRNAs. Our work reveals a potential direct role for APOBEC3A in ribosome biogenesis likely independent of its editing function. More broadly, we found an additional function of APOBEC3A in cancer pathology through its function in ribosome biogenesis, expanding its relevance as a target for cancer therapeutics.

## Introduction

Ribosome biogenesis, while a universal process in all living organisms, has increased complexity in human cells. Eukaryotic ribosome biogenesis starts in the nucleolus, a membraneless subnuclear compartment, with the transcription of the polycistronic 47S pre-ribosomal RNA

PRJNA935922). All underlying numerical values for all figures and supporting information figures are reported in S2 Data. All original, uncropped and minimally adjusted blot and gel images are reported in a supporting information PDF file (S1 Raw Images).

**Funding:** National Institutes of Health (NIH), nih. gov [1R35GM131687 to S.J.B., 1F31DE030332 to M.A.M., T32GM007223 to C.J.B., M.A.M., S.J.B., 1S10OD030363-01A1 to Yale Center for Genome Analysis] The funders did not play a role in study design, data collection, analysis, decision to publish, or preparation of the manuscript.

**Competing interests:** The authors have declared that no competing interests exist.

**Abbreviations:** 5-EU, 5-ethynyl uridine; AICDA, activation induced cytidine deaminase; APOBEC, apolipoprotein B mRNA editing catalytic polypeptide-like; co-IP, co-immunoprecipitation; COSMIC, Catalogue of Somatic Mutations in Cancer; ddPCR, digital droplet PCR; ETS, external transcribed spacer; EV, empty vector; BL, fibrillarin; FBS, fetal bovine serum; GTEx, Genotype-Tissue Expression; IF, immunofluorescence; ITS, internal transcribed spacer; LSU, large 60S subunit; pre-rRNA, pre-ribosomal RNA; RAMP, ratio analysis of multiple precursors; RB, ribosome biogenesis; rDNA, ribosomal DNA; RNAP1/3, RNA polymerase 1 / 3; RSEM, RNA-seq by Expectation-Maximization; SD, standard deviation; siNT, non-targeting siRNA; SNV, single-nucleotide variant; ssDNA, singlestranded DNA; SSU, small 40S subunit; TCGA, The Cancer Genome Atlas

(pre-rRNA) by RNA polymerase I (RNAP1). The pre-rRNA goes through a progression of processing, modification, and maturation steps with the help of transient assembly factors. Pre-rRNA assembly with ribosomal proteins yields the mature small 40S subunit (SSU) (18S rRNA) and large 60S subunit (LSU) [28S, 5.8S, and RNA polymerase 3 (RNAP3) transcribed 5S rRNA] [1,2]. While much early work focused on understanding ribosome biogenesis in baker's yeast, *Saccharomyces cerevisiae* [3], recent work has pivoted to revealing the more nuanced regulatory mechanisms that have evolved in humans and other vertebrate species [2,4]. In particular, several screens to identify novel ribosome biogenesis factors in human cells have been completed by our laboratory [5,6] and others [7–10]. While these screens do not establish precise mechanisms for all of the novel factors, they have unlocked new avenues of inquiry.

An up-regulation of ribosome biogenesis and protein synthesis is tightly associated with cancer (reviewed in [11,12]). The nucleolus is pleiotropic itself [13], as are various ribosome biogenesis factors with dual-functions in other cellular processes [14–16]. A main connection to the nucleolus is cell cycle progression through the TP53-mediated nucleolar stress response [17–19]. This stress response can result in nucleolar structure alterations, which is also an attribute of cancer cells; however, cancer cells exhibit increased number and size of nucleoli, correlating with greater ribosome production [20,21]. The reliance that proliferating cancer cells have on increased number of ribosomes is exemplified by the fact there are multiple cancer therapeutics in various stages of development that target the ribosome biogenesis pathway directly [22,23].

While cell growth in cancer relies on increased ribosome biogenesis, carcinogenesis is driven by somatic genomic DNA damage and mutation [24]. One somewhat newly implicated contributing factor to genomic instability is the aberrant function of the cytidine deaminase family of proteins [25,26]. Cytidine deaminases convert cytosine to thymine in single-stranded DNA (ssDNA) or to uracil in RNA, and have established biological roles in the immune response through hypermutation of the antibody variable region [activation induced cytidine deaminase (AICDA)] [27,28] and mutation of infecting viral genomes and endogenous retro-viral elements [apolipoprotein B mRNA editing catalytic polypeptide-like 3 subfamily members (APOBEC3s)] [29,30]. The first studied APOBEC, APOBEC1 [31,32], edits *apoB* mRNA and other RNAs [33], while still maintaining activity towards ssDNA [34]. On the other hand, APOBEC2 and APOBEC4 have no observed in vitro catalytic activity [34,35]. These useful editing functions can have unintended and deleterious consequences when the host genome is targeted. APOBEC3A and APOBEC3B expression is detected in many cancer types and their own classified Catalogue of Somatic Mutations in Cancer (COSMIC) mutational signatures [single base substation signatures (SBS) 2 and 13] are present in over 50% of human cancer types [36–38]. To date, APOBEC3A is emerging as the more prominent cytidine deaminase that edits the genome in cancer [39–41].

On top of genomic mutations, RNA editing is arising as another contributing element to the molecular pathogenesis of cancer [42]. Largely studied in an immune cell context, APOBEC3A [43–46] and APOBEC3G [47,48] possess mRNA editing activity. Similar to its increased expression resulting in off-target genomic DNA editing, APOBEC3A has been shown to edit mRNAs when expressed in cancer [49] and this is possibly true for other APOBEC3 family members as well [50]. While not completely understood, APOBEC3A is a multi-faceted DNA/RNA editing enzyme that can provide a defense against viruses, but its non-discriminatory editing also contributes to cancer disease progression through aberrant editing of the genome and potentially of mRNAs.

Here, we provide compelling evidence that APOBEC3A regulates ribosome biogenesis in human cells, another distinct pathway that drives cancer. In a prior genome-wide siRNA

screen conducted by our laboratory, APOBEC3A and APOBEC4 were identified as regulators of nucleolar function [5]. While we confirmed the screening results for both APOBEC3A and APOBEC4, we directed our focus toward APOBEC3A for further investigation. Through a series of depletion and overexpression experiments, we establish a correlation between APO-BEC3A levels and cell cycle progression, cell growth, and protein synthesis. More precisely, we show that APOBEC3A is required for LSU maturation and pre-LSU rRNA processing. We demonstrate this role is likely through direct association with pre-ribosomal factors in the nucleolus. Notably, by introduction of a catalytically dead mutation in APOBEC3A, we show that its editing function is not required to increase protein synthesis and cell growth. Neverthe-less, we investigate how APOBEC3A editing and/or RNA binding regulates ribosome biogene-sis and the cell cycle through a novel nuclear RNA-sequencing experiment, identifying predicted sites of APOBEC3A C-to-U binding/editing on the pre-LSU rRNA and pre-mRNAs encoding nucleolar proteins and cell cycle regulators. Our results point towards the possibility that APOBEC3A is a direct and indirect modulator of ribosome synthesis, linking its expres-sion to cancer cell proliferation through a novel mechanism.

## Results

### Cytidine deaminases are up-regulated and correlate with a decrease in cancer survival

Due to the various connections between cytidine deaminases and cancer, we analyzed all 11 known human cytidine deaminases' mRNA expression levels across cancer types using Geno-type-Tissue Expression (GTEx) unmatched normal and The Cancer Genome Atlas (TCGA) matched normal and tumor samples [51]. We observed that 9 out of 11 cytidine deaminases have significantly higher expression in tumor versus normal tissue (S1 Fig). Since this does not provide any evidence for a functional role within the context of tumorigenesis or cancer progression, we sought to examine any correlations between cytidine deaminase expression and patient survival across cancer type using the associated patient survival data [51]. After stratifying expression into either high or low based on the mean expression level, 8 out of the 11 cytidine deaminases exam-ined had high expression levels associated with a significant reduction in patient survival (S2A Fig). Based on these 2 analyses, 7 out of the 11 cytidine deaminases are more highly expressed in cancer versus normal tissue and their higher expression is associated with decreased survival prob-ability (S2B Fig). We aimed to build upon these correlations with analysis of cytidine deaminase function in cancer by interrogating this protein family for yet to be discovered cellular roles.

### The cytidine deaminase, APOBEC3A, is a strong candidate regulator of ribosome biogenesis

We hypothesized that acute depletion of cytidine deaminases would reveal if any have a can-cer-driving function beyond known roles in DNA mutagenesis, which occurs over the course of many cell divisions [40,52]. One hallmark of cancer cells is the up-regulation of ribosome biogenesis, leading to increased cell growth and proliferation [53]. Therefore, we examined our laboratory's previously published genome-wide siRNA screen datasets [5,6] to identify cytidine deaminases that possibly regulate nucleolar function. Briefly, MCF10A cells normally have 2 to 3 nucleoli per cell nucleus [54]. However, depletion of ribosome biogenesis factors can alter this number, either increasing the number of cells harboring 1 nucleolus [5] or increasing the number of cells with 5 or more nucleoli [6]. We quantify this as a "one-nucleo-lus percent effect" where the average number of nucleoli in a given treatment is compared to the negative control non-targeting siRNA (siNT) which is set to 0% and the positive control

(siUTP4 in previously published screen [5] or siNOL11 in updated rescreen) which is set to 100%. In the previously published screens from our laboratory, out of the 10 cytidine deaminases analyzed, we found that only APOBEC3A and APOBEC4 were hits. Their siRNA depletion produced a one-nucleolus percent effect even higher than the positive controls siUTP4 and siNOL11, while the other cytidine deaminases did not meet this threshold (Fig 1B) [5,55]. As expected, this result correlated with a decrease in cell viability, both of which were remarkably lower than the total screen median of 83.7% compared to non-targeting siRNA (siNT) negative control (100%) (Fig 1B) [5]. These results indicate that APOBEC3A and APOBEC4 are potentially involved in making ribosomes, while it is less likely that other cytidine deaminase family members possess a function in ribosome biogenesis.

Furthermore, we took advantage of other available published ribosome biogenesis screening data to provide more insight on APOBEC3A, APOBEC4, and other cytidine deaminases' functions [6–10]. However, while some possess potential roles in 40S biogenesis [8], no cytidine deaminases met the threshold required to be deemed a hit in any of these other ribosome biogenesis screens (Data A in S1 Data). Thus, we did not include any other cytidine deaminases in our downstream analyses.

To validate our screening results, we repeated our established nucleolar number assay in triplicate with an updated siRNA technology (siON-TARGET, Horizon Discovery, Data B in S1 Data) which reduces off-target effects [59]. We observed a significant increase in cells harboring 1 nucleolus after siAPOBEC3A depletion (46.3% one-nucleolus cells, 224.8% effect), but only a modest increase in one-nucleolus harboring cells after siAPOBEC4 depletion (23.9% one-nucleolus cells, 33.6% effect) compared to the negative control siNT (19.2% one-nucleolus cells, 0% effect) and positive control siNOL11 (29.2% one-nucleolus cells, 100% effect) treatments (Fig 1C and Data C in S1 Data). Furthermore, we deconvoluted the siON-TARGET pool by testing each of the 4 individual siRNA's ability to produce the one-nucleolus phenotype (Data D in S1 Data). We used a stringent >3 SD from the negative control (siNT) as a cutoff for the one-nucleolus phenotype for both the pool and individual siRNA treatments. For siAPOBEC3A treatment, both the pool and 2 out of 4 individual siRNAs passed the cutoff for reducing nucleolar number. While the average of the siAPOBEC4 pool and 2 out of 4 individual siRNAs surpassed the 3 SD cutoff, only 1 of the 3 siAPOBEC4 replicates surpassed this cutoff (Fig 1D, top). Again, decreases in nucleolar number correlated with concomitate decreases in cell viability (Fig 1D, bottom). Due to the stronger and more consistent results with siAPOBEC3A compared to siAPOBEC4, we decided to focus solely on APOBEC3A as the more promising novel ribosome biogenesis factor.

Additionally, we validated APOBEC3A depletion by western blotting using an antibody that detects both APOBEC3A and 3B. We observed a significant reduction in APOBEC3A levels, but not APOBEC3B levels after treatment with the APOBEC3A-specific siRNA pool and the 2 individual siRNAs that passed deconvolution (siRNAs #1 and #2) (Figs 1E and S3A). APOBEC3A mRNA levels are typically very low in most cell types and in some cases undetectable by qRT-PCR [60,61]. Consequently, we were unable to detect *APOBEC3A* mRNA by qRT-PCR in MCF10A cells but did successfully observe depletion of *APOBEC3A* mRNA levels by digital droplet PCR (ddPCR) after siAPOBEC3A pool treatment (Fig 1F). Based on these preliminary results, we considered APOBEC3A to be a strong candidate ribosome biogenesis factor worthy of further investigation.

## APOBEC3A is required for cell cycle progression and protein synthesis

Because APOBEC3A's acute depletion for 72 h led to reduced cell viability, we predicted that it would also result in the dysregulation of cell cycle progression. Ribosome biogenesis and the

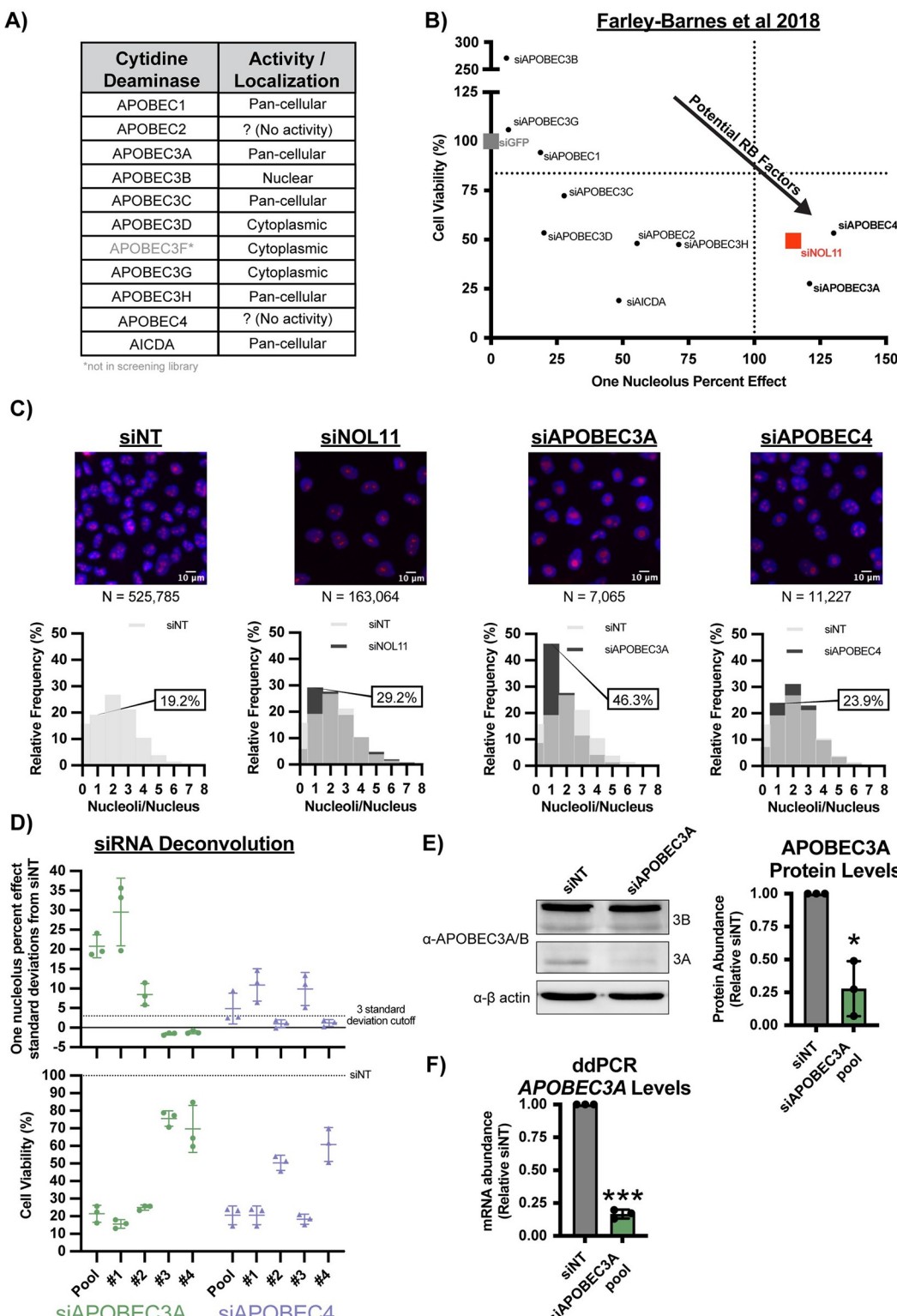

**Fig 1. The cytidine deaminases, APOBEC3A and APOBEC4, are novel ribosome biogenesis factors in MCF10A cells.**
(A) The cytidine deaminase family of proteins has diverse activities and subcellular localizations. This table lists all 11 known human cytidine deaminases with their subcellular localizations or if they do not have known catalytic activity. This information regarding cytidine deaminase function was curated from the following sources: [56–58]. (B) The cytidine deaminases APOBEC3A and APOBEC4 were hits in our laboratory's previous genome-wide siRNA screen for ribosome biogenesis regulators in MCF10A cells [5]. Graph of data on cell viability (y-axis) and one-nucleus harboring cells (one-

nucleolus percent effect, x-axis) from 72 h siRNA treatment targeting 10 of 11 known human cytidine deaminases. siGFP (gray) was a negative control (100% viability, 0% one-nucleolus percent effect) and siUTP4 (100% one-nucleolus percent effect, vertical dotted line) and siNOL11 (red) were positive controls. The entire screen cell viability median was 83.7% (horizontal dotted line). siAPOBEC3A and siAPOBEC4 treatments (bold, lower right quadrant) passed the thresholds for both one-nucleolus percent effect and reduced cell viability. (C) APOBEC3A and APOBEC4 depletion by siON-TARGET pools reduces nucleolar number in MCF10A cells. (Top) Representative merged images of nuclei stained with Hoechst (blue) and nucleoli stained with α-fibrillarin (magenta), N = number of nuclei (cells) analyzed. Non-targeting siRNA treatment (siNT) was used as a negative control (2–3 nucleoli per nucleus) and siNOL11 was used as a positive control (reduction in nucleolar number). (Bottom) Histograms of relative frequency of nucleoli per nucleus. siNT (light gray), siNOL11, siAPOBEC3A, and siAPOBEC4 (dark gray), overlap between siNT and other siRNA treatments (intermediate gray). Percentage of one-nucleolus harboring cells is indicated for each siRNA treatment. (D) siRNA deconvolution for siAPOBEC3A and siAPOBEC4 siON-TARGET pools. (Top) siAPOBEC3A and siAPOBEC4 pool and 2 out of 4 individual siRNA treatments significantly reduce nucleolar number in MCF10A cells. One-nucleolus percent effect set relative to standard deviations from negative control, siNT; 3-standard deviation from negative control cutoff (horizontal dashed-line). Three biological replicates plotted mean ± SD. (Bottom) The same pool and individual siRNA treatments in (Top) that reduce nucleolar number reduce cell viability. Percent cell viability set relative to siNT treatment (100%, horizontal dashed line). Three biological replicates plotted mean ± SD. (E) Western blots confirming APOBEC3A protein depletion after siAPOBEC3A pool treatment. (Left) Representative western blot using an α-APOBEC3A/B antibody. siNT is a negative control. α-β-actin is shown as a loading control. (Right) Quantification of APOBEC3A protein levels normalized to β-actin signal and relative to siNT negative control. Three biological replicates plotted mean ± SD. Data were analyzed by Student's t test, * $p \leq 0.05$. (F) ddPCR confirming APOBEC3A mRNA depletion after siAPOBEC3A pool treatment. ddPCR measuring APOBEC3A mRNA levels normalized to ACTB internal control and relative to siNT negative control. Three technical replicates of 3 biological replicates, plotted mean ± SD. Data were analyzed by Student's t test, *** $p \leq 0.001$. All underlying numerical values for figure found in S2 Data. ddPCR, digital droplet PCR; RB, ribosome biogenesis; SD, standard deviation.

cell cycle are interconnected [62,63], with dysregulation of both having consequences in cancer pathogenesis [64]. Using images collected from the screening assay, we quantified DNA content from Hoechst staining of siNT and siAPOBEC3A treated MCF10A cells as in [6,65]. APOBEC3A siRNA depleted cells displayed a sharp increase in S-phase cells and an accompanying decrease in sub-G1 and G1 phase cells compared to siNT treated cells (Fig 2A).

To connect APOBEC3A's potential role in ribosome biogenesis with the cell cycle, we tested for the induction of the nucleolar stress response [17–19] after APOBEC3A siRNA depletion. The nucleolar stress response results in a change in nucleolar morphology, increased levels of TP53 and its transcriptional target CDKN1A [66], and ultimately cell cycle arrest and apoptosis. We tested for an induced nucleolar stress response by measuring TP53 and CDKN1A protein levels by western blotting after siAPOBEC3A treatment in MCF10A cells, which have wild-type p53 [67,68]. We observed both a significant increase in TP53 and CDKN1A levels in siAPOBEC3A treated cells compared to siNT treated cells (Fig 2B). These results agree with a genome-wide RNAi screen in A549 cells to identify proteins whose depletion leads to TP53 stabilization, where APOBEC3A was a top hit (Data A in S1 Data) [19]. Taken together, these results indicate that acute depletion of APOBEC3A induces the nucleolar stress response that leads to inhibition of cell cycle progression and to reduced cell viability.

Since the nucleolar stress response (stabilization of TP53) results from inhibiting nucleolar ribosome biogenesis [17–19], we predicted that APOBEC3A siRNA depletion would result in a reduced pool of functional ribosomes, and thus decreased protein synthesis. We used a puromycin incorporation assay [69] to measure changes in global protein synthesis upon siAPOBEC3A treatment of MCF10A cells. After siRNA depletion, 1 μM puromycin was added, which is incorporated into nascent polypeptides, for 1 h. We measured puromycin incorporation over this time period by western blotting with α-puromycin antibodies and observed a significant decrease in global protein synthesis after APOBEC3A depletion. The reduction in protein synthesis upon APOBEC3A depletion was to a similar extent as depletion of the positive control, the LSU ribosomal protein RPL4 (uL4) (Fig 2C). Furthermore, this result was recapitulated using both individual siRNAs that target APOBEC3A (#1 and #2), which both

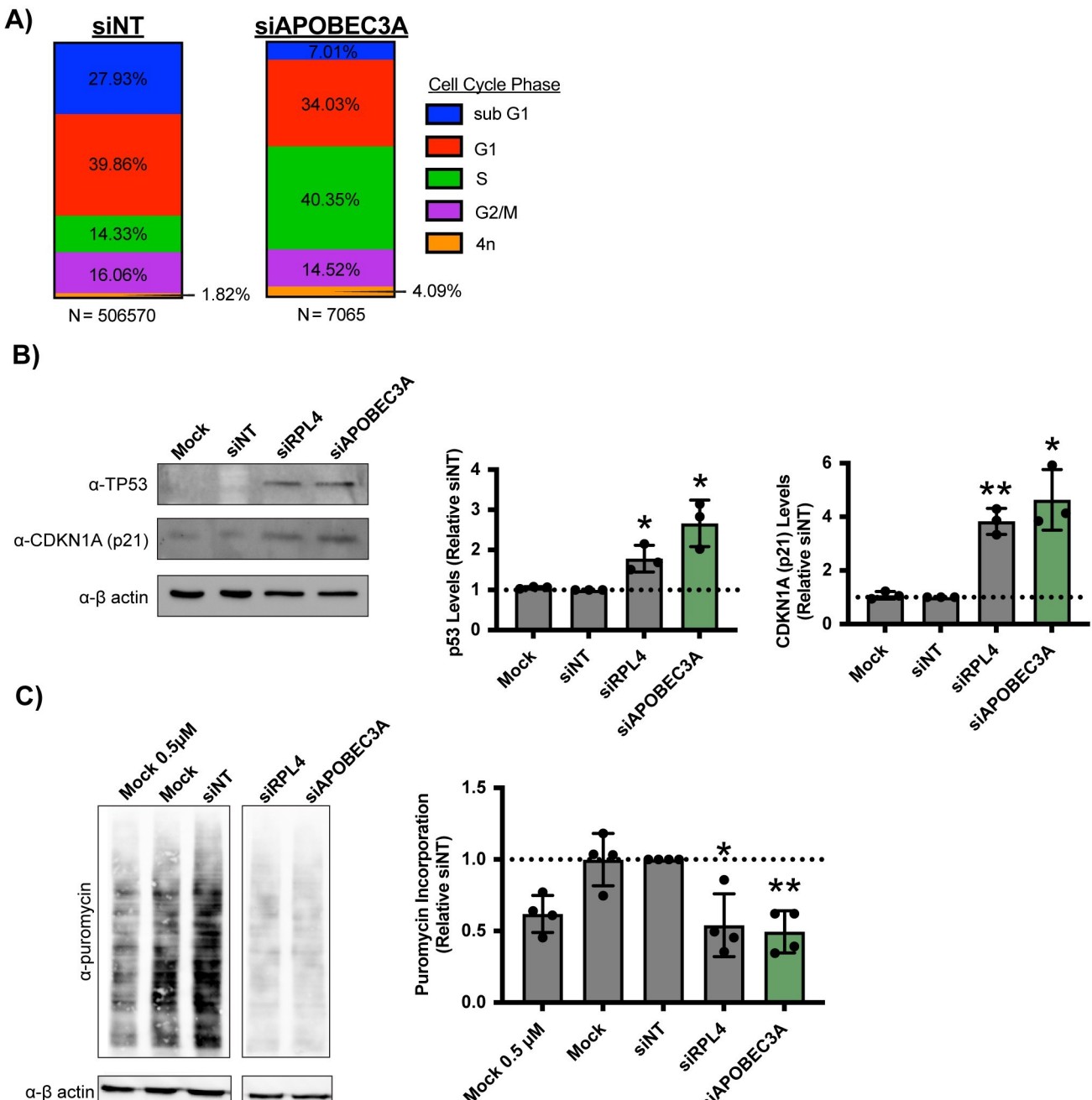

**Fig 2. APOBEC3A is required for cell cycle progression and global protein synthesis in MCF10A cells.** (A) siAPOBEC3A depletion (siRNA pool) leads to changes in cell cycle distribution in MCF10A cells. DNA intensity was measured by Hoechst staining (from images in Fig 1C) and the $\text{Log}_2$ DNA intensities were normalized to siNT negative control where its G1 peak = 1.0 and G2 peak = 2.0. Cell cycle phases were determined by the following normalized $\text{Log}_2$ integrated intensities. Sub G1 < 0.75, G1 = 0.75–1.26, S = 1.25–1.75, G2/M = 1.75–2.5, >4n > 2.5. Data was plotted as relative proportion of cells within each phase indicated by color. $N$ = number of cells pooled across 3 replicates. (B) siAPOBEC3A depletion (pool) induces the nucleolar stress response in MCF10A cells. (Left) Representative western blot using α-TP53 and α-CDKN1A (p21) antibodies. Mock and siNT are negative controls. α-β-actin is shown as a loading control. (Right) Quantification of TP53 and CDKN1A (p21) protein levels normalized to α-β-actin signal and relative to siNT negative control. Three biological replicates plotted mean ± SD. Data were analyzed by Student's $t$ test, ** $p \leq 0.01$, * $p \leq 0.05$. (C) siAPOBEC3A depletion (pool) reduces global protein synthesis in MCF10A cells. After 72 h siRNA depletion, 1 μM puromycin was added for 1 h to measure translation. (Left) Representative western blot using an α-puromycin antibody. Mock and siNT are negative controls, siRPL4 is a positive control, and Mock 0.5 μM is a control to indicate robust quantification. α-β-actin is shown as a loading control. (Right) Quantification of puromycin signal normalized to β-actin signal and relative to siNT negative control. Four biological replicates plotted mean ± SD. Data were analyzed by one-way ANOVA with Dunnett's multiple comparisons test, ** $p \leq 0.01$, * $p \leq 0.05$. All underlying numerical values for figure found in S2 Data. SD, standard deviation.

resulted in a significant reduction in protein synthesis when transfected into MCF10A cells (S3B Fig).

Ribosome biogenesis is required across cell types. Therefore, we expected that APOBE-C3A's function in making ribosomes would be conserved to other cell types as well. We confirmed APOBEC3A's overall function in making ribosomes in HeLa cells. Because the MCF10A cell line is not a cancer cell line, HeLa cells provide us with a more direct test of APOBEC3A's role in a cancer context. Moreover, APOBEC3A has been shown to be a probable driver of cervical cancer pathogenesis (reviewed in [58,70]). As we observed in MCF10A cells, we were able to confirm both decreased APOBEC3A protein levels and global protein synthesis after siAPOBEC3A treatment compared to siNT in HeLa cells (S4A and S4B Fig). In sum, our results show that APOBEC3A is required for cell cycle progression, most likely due in part to its depletion leading to the nucleolar stress response and decreased translation.

## APOBEC3A is required for nucleolar rRNA biogenesis by regulating steps downstream of rRNA transcription

To gain insight into APOBEC3A's role in making ribosomes, we utilized an assay we previously developed in our laboratory to measure nascent nucleolar rRNA biogenesis [71]. Briefly, after siRNA knockdown, MCF10A cells were treated with 5-ethynyl uridine (5-EU) for 1 h followed by biocompatible click chemistry to visualize the 5-EU signal. By co-staining with the nucleolar marker fibrillarin (FBL), we can then measure the 5-EU signal residing within the nucleolus. This readout combines both rRNA transcription and the stability of nucleolar (pre-)rRNA. We found that siAPOBEC3A treated cells displayed a modest decrease in nucleolar 5-EU signal, corresponding to a 50.8% nucleolar rRNA biogenesis percent inhibition relative to the negative control siNT (0% inhibition) and positive control siPOLR1A (the largest subunit of RNAP1, 100% inhibition) (S5A and S5B Fig). Based on our previously published results testing established ribosome biogenesis factors using this assay [71], depletion of factors required for pre-rRNA transcription or both pre-rRNA transcription and processing result in higher nucleolar rRNA biogenesis percent inhibition values of greater than approximately 80%, while factors only required pre-rRNA maturation result in lower percent inhibition values approximately 50% to 80%, and factors not involved in making ribosomes at all have a percent inhibition value less than 50% [71]. Thus, it is likely that APOBEC3A is not required for pre-rRNA transcription but plays a role somewhere downstream in the ribosome biogenesis pathway.

We utilized multiple assays that are more direct readouts of pre-rRNA transcription to confirm that APOBEC3A is not required for the first step in making ribosomes. In one approach, we used qRT-PCR to measure steady state levels of the primary pre-rRNA transcript in MCF10A cells depleted of APOBEC3A. While RNAP1 transcribes the 47S pre-rRNA precursor, immediate cleavage at site A' yields the more readably detectable 45S pre-rRNA precursor, so this assay is measuring both 47S and 45S levels (S5C Fig) [72,73]. siAPOBEC3A treatment did not significantly alter 47S+45S pre-rRNA levels compared to the negative control siNT treatment; however, the positive control siPOLR1A treatment did significantly reduce these levels (S5D Fig).

As another readout of pre-rRNA transcription, we used a dual-luciferase reporter [74] to test for changes in rDNA promoter activity upon APOBEC3A depletion, which we have successfully used previously [5,6,75]. After siRNA depletion, the firefly luciferase reporter, under control of the rDNA promoter (−410 to +327), is measured relative to a constitutive *Renilla* luciferase control 24 h post plasmid transfection (S5E Fig) [74]. siAPOBEC3A treatment did not significantly reduce rDNA promoter activity compared to siNT treatment, while the

positive control, siPOLR1A treatment, did significantly reduce rDNA promoter activity (S5F Fig). Collectively, APOBEC3A is likely required for ribosome biogenesis during step(s) down stream of pre-rRNA transcription.

## APOBEC3A is required for pre-LSU rRNA processing and maturation

To reveal if one ribosomal subunit was preferentially affected by APOBEC3A siRNA depletion, we quantified the levels of mature 18S (SSU) and 28S (LSU) rRNAs on an Agilent Bioanalyzer. We used total RNA to quantify both a 28S/18S rRNA ratio and overall rRNA levels in siAPO-BEC3A treated MCF10A cells. MCF10A cells normally have a 28S/18S ratio of approximately 2 in siNT treated cells, which was significantly decreased 2-fold after APOBEC3A siRNA depletion (Fig 3A, left). This decreased ratio can be attributed to reduced 28S rRNA levels specifically and not to changes in 18S rRNA levels (Fig 3A, right). The observed reduction in 28S rRNA levels points to APOBEC3A playing a role in pre-LSU rRNA processing and/or maturation.

We also harvested nuclear RNA from MCF10A cells to test if siAPOBEC3A-mediated 28S rRNA maturation defects occur before or after ribosome subunit export out of the nucleus. In contrast, Bioanalyzer analysis of total RNA picks up almost exclusively mature rRNAs, which are in the cytoplasm (Fig 3B, left). As we had anticipated, the nuclear Bioanalyzer traces have a much greater proportion of large sized RNAs detected around the 18S and 28S mature rRNA peaks compared to total RNA analysis. Presumably, many of these correspond to pre-rRNA intermediate species (Fig 3B and 3C). We quantified these results by taking the 28S/18S rRNA ratio which, similar to the results with total RNA (Fig 3A), was reduced again approximately 2-fold in APOBEC3A depleted MCF10A cells compared to siNT treated cells (Fig 3D, left). Therefore, siAPOBEC3A treatment of MCF10A cells results in an LSU-specific maturation defect within the nucleus. Aside from the expected changes in 28S abundance, one of the most striking differences between the siNT and siAPOBEC3A nuclear RNA Bioanalyzer traces was an increase in the abundance of an RNA species around approximately 6,500 nucleotides in size. This corresponds to the size of the 32S pre-rRNA intermediate, a precursor to the mature 28S and 5.8S LSU rRNAs (Fig 3E) [77]. We quantified the 28S/32S ratio and observed approximately 2-fold decrease in APOBEC3A depleted cells compared to the siNT negative control (Fig 3D, right). This decreased ratio indicates a pre-rRNA processing defect visualized by an aberrant build-up of the 32S pre-rRNA and a subsequent decrease in mature 28S rRNA levels (Fig 3E), which we confirmed using northern blotting.

We performed northern blotting for the pre-rRNAs to substantiate our claim that pre-28S processing defects were occurring after APOBEC3A siRNA depletion in MCF10A cells. We used an internal transcribed spacer sequence 2 (ITS2) probe to measure steady-state levels of the pre-rRNA precursors leading to the maturation of the 5.8S and 28S rRNAs (Fig 3E). We observed a noticeable increase in the 32S pre-rRNA intermediate and corresponding decrease in the downstream 12S pre-rRNA in siAPOBEC3A compared to siNT treatment (Fig 3F). These results were quantified using ratio analysis of multiple precursors (RAMP) [76], which confirmed a significant increase in the 32S precursor and a decrease in the 12S precursor (product of 32S) compared to all of their upstream precursors in the LSU processing pathway (Fig 3E and 3G). Agreeing with our Bioanalyzer results, methylene blue staining also indicated a decrease in mature 28S/18S rRNA ratio after APOBEC3A siRNA depletion (Fig 3H).

To investigate a potential mechanism underlying APOBEC3A's regulation of pre-28S rRNA processing, we tested for changes in the protein levels of the ITS2 nuclease, LAS1L [78,79], after APOBEC3A siRNA depletion in MCF10A cells. We did not observe any significant changes in LAS1L levels, ruling out regulation through LAS1L as a potential mechanism

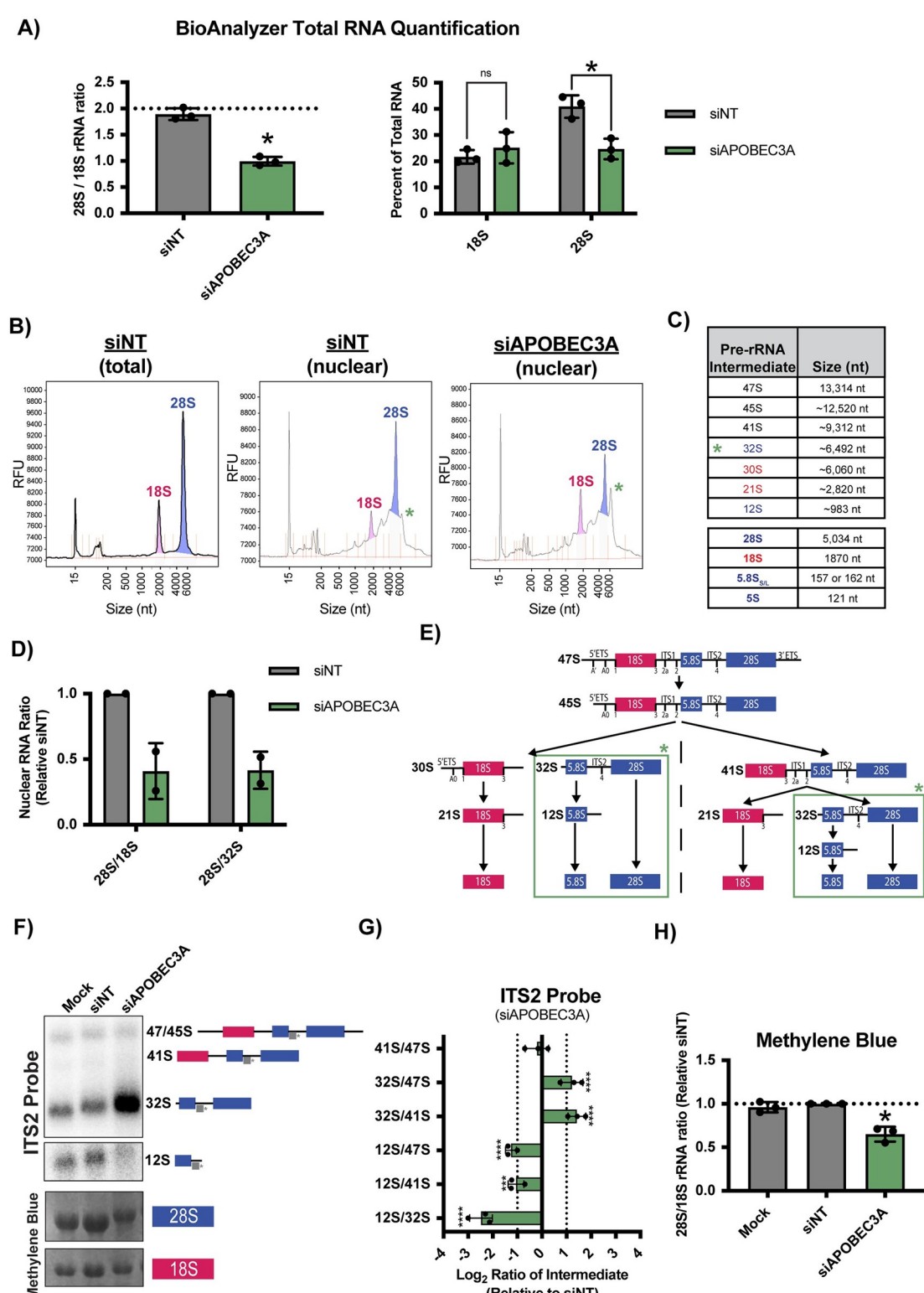

**Fig 3. APOBEC3A is required for pre-LSU processing prior to nuclear export.** (A) siAPOBEC3A depletion (pool) reduces mature 28S rRNA levels in MCF10A cells. Agilent Bioanalyzer total RNA quantification. (Left) 28S/18S mature rRNA ratio with siNT as a negative control. Expected normal 28S/18S rRNA ratio of approximately 2.0 (dashed horizontal line). (Right) Percent of total RNA levels for 18S and 28S rRNA with siNT as a negative control. Three biological replicates plotted mean ± SD. Data were analyzed by Student's *t* test, ** $p \leq 0.01$. (B) Nuclear RNA is enriched for the pre-rRNAs and siAPOBEC3A depletion (pool)

increases the abundance of approximately 6.5 kb RNA species (32S pre-rRNA) in the nucleus of MCF10A cells. Representative Agilent Bioanalyzer traces from total cell RNA (left) and nuclear fraction RNA (middle, right). Green star indicates an apparent nuclear RNA species (approximately 6.5 kb) with increased abundance upon siAPOBEC3A treatment compared to the siNT negative control. (C) Table of human pre-rRNA intermediate and mature rRNA species and their approximate length in nucleotides (nt). Green star indicates the 32S pre-rRNA is increased after siAPOBEC3A depletion. (D) siAPOBEC3A depletion (pool) reduces nuclear 28S/18S and 28S/32S pre-rRNA ratios. Quantification of nuclear RNA Agilent Bioanalyzer traces. pre-rRNA ratios are reported relative to the siNT negative control. Two biological replicates are plotted mean ± SD. (E) Diagram of pre-rRNA processing in humans. The ETS and ITS sequences are removed to yield the mature 18S, 5.8S, and 28S rRNAs. The green box indicates the expected defect in processing after APOBEC3A depletion, where ITS2 processing is reduced leading to an increase in 32S levels and a decrease in 12S levels. (F, G) siAPOBEC3A depletion (pool) inhibits pre-LSU rRNA processing in ITS2 to reduce LSU maturation. (F) (Top) Processing of the 32S to the 12S pre-rRNA is reduced after siAPOBEC3A depletion. Representative northern blot using an ITS2 probe to measure steady-state levels of pre-rRNAs that lead to the formation of the mature 5.8S and 28S rRNAs (LSU). The detected pre-rRNAs are indicated along with the ITS2 probe (gray). (Bottom) Methylene blue staining of the mature 28S and 18S rRNAs as a loading control. Mock transfected and siNT are negative controls. (G) Quantification of defective pre-rRNA processing results in (F). RAMP [76] quantification of northern blotting using an ITS2 probe in (F). Log$_2$ fold change of pre-rRNA ratios were quantified relative to siNT negative control. Three biological replicates plotted mean ± SD. Data were analyzed by two-way ANOVA, **** $p \leq 0.0001$, *** $\leq 0.001$. (H) Quantification of mature rRNA levels in (F). Methylene blue staining quantification of mature 28S to 18S rRNA ratio relative to siNT negative control. Three biological replicates plotted mean ± SD. Data were analyzed by Student's $t$ test, * $p \leq 0.05$. All underlying numerical values for figure found in S2 Data. ETS, external transcribed spacer; ITS, internal transcribed spacer; RAMP, ratio analysis of multiple precursors; SD, standard deviation.

(S6 Fig). Taken together, our results demonstrate that APOBEC3A is required for pre-LSU rRNA processing at ITS2 cleavage that processes the 32S to 12S pre-rRNA prior to nuclear export in MCF10A cells.

## A subset of APOBEC3A is found within the nucleolus and associates with critical pre-ribosome factors

While we have observed that APOBEC3A is required for pre-LSU biogenesis, our results do not demonstrate whether APOBEC3A acts directly within the nucleolus. Previous studies have shown that APOBEC3A resides throughout the cell, including in the nucleus [56–58], but no work has focused on whether APOBEC3A is nucleolar. Because of the lack of the commercial availability of specific antibodies to APOBEC3A that do not also detect other APOBEC family members [80], and because endogenous levels are normally low [49,81], we turned to using a tagged APOBEC3A overexpression approach. We transfected a C-terminal 3X-FLAG tagged version of APOBEC3A to detect its subcellular localization in HeLa cells using immunofluorescence (IF) and co-immunoprecipitation (co-IP). For these experiments, we chose HeLa cells due to their ease of transfection (S7 Fig) and because of their relevance as a cancer cell line.

To test the subcellular localization of the tagged APOBEC3A, we performed IF 24 h post transfection. In addition to staining with an α-FLAG antibody, we co-stained cells with DAPI (nuclei) and α-FBL (fibrillarin; nucleoli). We observed that about 40% of cells were positive for FLAG staining and we subsequently selected them for colocalization analysis (Fig 4A). Consistent with previous findings, APOBEC3A-FLAG showed pan-cellular localization. More specifically, colocalization analysis revealed APOBEC3A-FLAG was moderately enriched in the nucleus (DAPI, average Pearson correlation coefficient = 0.62) and colocalized with the nucleolus (FBL, average Pearson correlation coefficient = 0.42) (Fig 4B). Therefore, a subset of APOBEC3A-FLAG localizes to the nucleolus, indicating that it is capable of directly regulating ribosome biogenesis.

To more closely examine the subset of APOBEC3A-FLAG within the nucleolus, we performed co-IPs followed by western blotting of essential nucleolar proteins [fibrillarin (FBL) and RPA194]. We were able to utilize a positive control by blotting for KAP1 (TRIM28), a protein that had previously been shown to associate with both APOBEC3A [82] by reciprocal co-

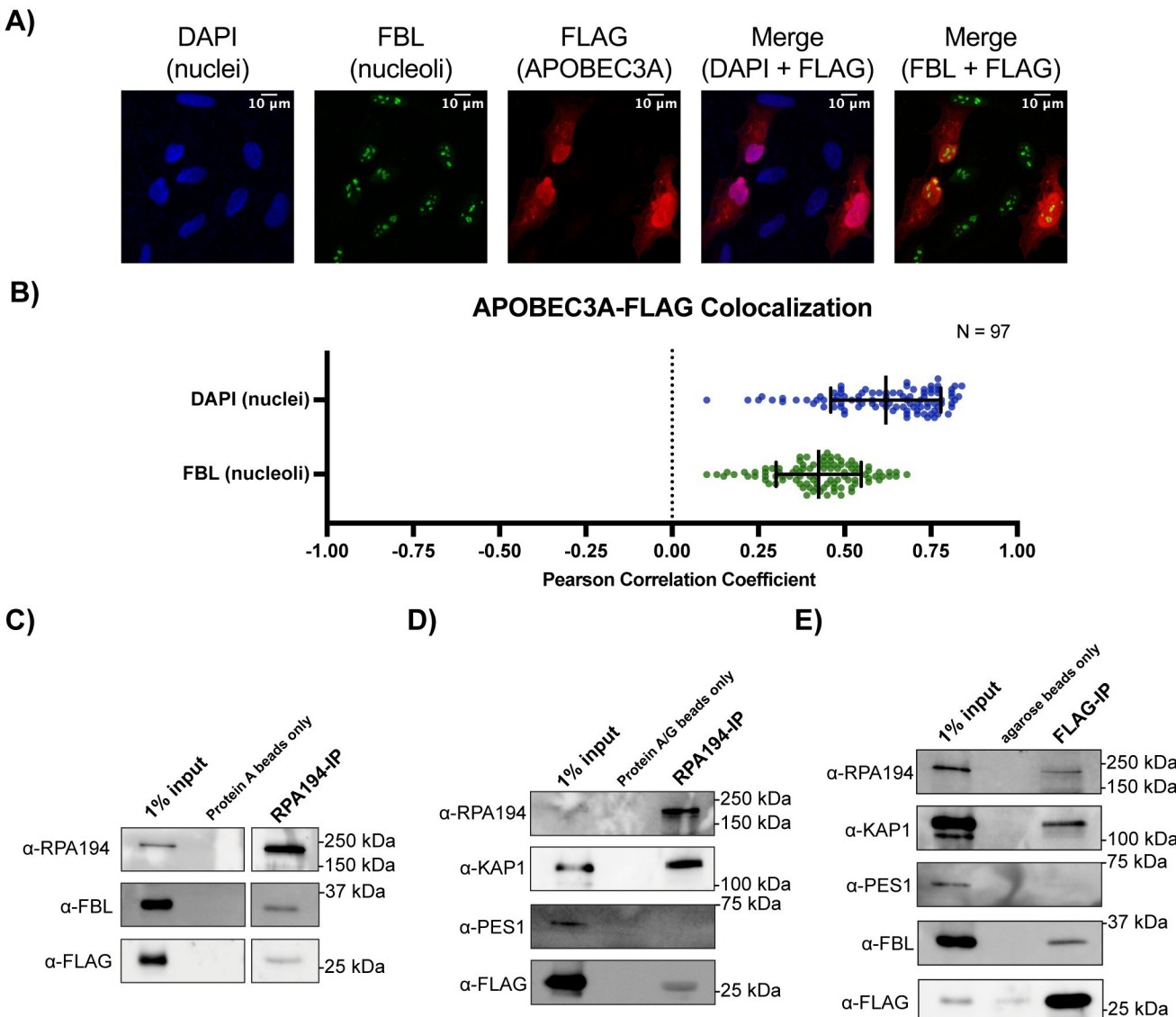

**Fig 4. A subset of APOBEC3A-FLAG localizes to the nucleolus and associates with RPA194 and FBL.** (A, B) APOBEC3A-FLAG colocalizes with nuclei and the nucleolar protein FBL. (A) Representative images of HeLa cells transfected for 24 h. Nuclei are stained with DAPI (blue), nucleoli detected with α-fibrillarin (green), and then stained with α-FLAG (red) for the transfected C-terminal 3X-FLAG APOBEC3A. Three (3) out of 9 cells were positive for expression of APOBEC3A-FLAG. Representative merged images showing DAPI and FBL colocalization with FLAG staining. (B) Pearson correlation coefficient of α-FLAG staining with DAPI and FBL quantification. $N = 97$ cells positive for FLAG staining plotted mean ± SD. (C–E) APOBEC3A-FLAG coimmunoprecipitates nucleolar proteins RPA194 and FBL. (C) HeLa whole cell extracts were immunoprecipitated with α-RPA194 antibody. Input corresponds to 1% of the whole cell extract used for immunoprecipitation. Unconjugated protein A beads only were used as a negative control. RPA194, FBL, and APOBEC3A-FLAG were detected by western blotting with the indicated antibody. Representative western blot images shown for 1 of 2 biological replicates. (D) Panel as in C, except with protein A/G beads only used as a negative control and RPA194, KAP1, PES1, and FLAG were detected by western blotting with the indicated antibody. (E) Panel as in C, except α-FLAG antibody was used and agarose beads only were used as a negative control. RPA194, KAP1, PES1, FBL, and APOBEC3A-FLAG were detected by western blotting. All underlying numerical values for figure found in S2 Data. FBL, fibrillarin; SD, standard deviation.

IPs and with RPA194 by KAP1 co-IP followed by mass spectrometry [83] and a negative control PES1 which we previously showed does not associate with RPA194 [75]. Since protein A/G beads produced background signal when blotting using a α-FBL antibody, we first show α-RPA194 conjugated, but not unconjugated protein A beads, co-IP endogenous FBL as

expected as well as transfected APOBEC3A-FLAG (Fig 4C). Similarly, α-RPA194 conjugated, but not unconjugated protein A/G beads, co-IP endogenous KAP1 and transfected APOBE-C3A-FLAG, but not endogenous PES1 (Fig 4D). Lastly, we performed reciprocal co-IPs using α-FLAG antibody conjugated agarose beads. α-FLAG conjugated, but not agarose beads alone, co-IP endogenous RPA194, KAP1, and FBL, but not PES1 (Fig 4E). These results suggest a direct role for APOBEC3A in ribosome biogenesis where a subset of APOBEC3A-FLAG resides within the nucleolus and associates either directly or in complex with key nucleolar proteins, including RPA194 and FBL.

## Transient overexpression of both wild-type and catalytically dead mutated (C106S) versions of APOBEC3A increase cell growth and protein synthesis

Since we have established using siRNA depletion that APOBEC3A is required for ribosome biogenesis and has the capability for a direct role within the nucleolus (Figs 1D and 2), we chose to probe if APOBEC3A overexpression would drive ribosome biogenesis, protein synthesis, and cell proliferation. We hypothesized that APOBEC3A overexpression would have the inverse effect that we observed with depletion. Critically, we asked: is APOBEC3A's editing activity required for its role in promoting cell growth and protein synthesis? To this end, we continued to utilize the APOBEC3A-FLAG transient expression system in HeLa cells. Because APOBEC3A-FLAG is expressed under the control of the CMV promoter, we are able to detect it where it would be difficult to detect the endogenous APOBEC3A. We tested not only wild type (3A WT), but also an established catalytically dead mutated form C106S (3A C106S) that does not possess editing capability [82,84,85] and a 3X-FLAG empty vector (EV) negative control (Fig 5A). We expected that if APOBEC3A editing is not required, then both 3A WT and the C106S mutated form will produce similar results in these experiments. In contrast, if editing is required then only 3A WT will produce positive results.

To test if transient APOBEC3A overexpression can drive cell proliferation, we examined changes in cell growth using the Cell Titer-Glo 2.0 assay 48 h post transfection using increasing amounts of plasmid. Overexpression of both 3A WT and 3A C016S increased cell growth compared to the EV control (Fig 5B). This positively correlated with the amount of overexpression, where higher amounts of transfected plasmid increased cell growth (Fig 5B). These results suggest that transient APOBEC3A overexpression can increase cell growth independent of its editing activity and thus also independent of its mutagenic role in cancer.

We utilized the same puromycin incorporation assay as in Fig 2C [69] to test if APOBEC3A overexpression also increases protein synthesis. After 48 h of overexpression, we observed an increase in puromycin signal with the higher 1.0 ng/μl plasmid concentration, but not with the lower 0.5 ng/μl concentration, for both 3A WT and 3A C106S, compared to the EV control (Fig 5C). For both 3A WT and 3A C106S, puromycin signal increased with the higher amount of overexpression (Fig 5C). Consistent with our cell growth results, higher APOBEC3A-FLAG overexpression, for both WT and C106S, increases protein synthesis. Surprisingly, we observed a trending increase in 3A C106S levels compared to 3A WT under the same transfection conditions (Fig 5D). Taken together, increasing overexpression of both wild-type and a catalytically dead version of APOBEC3A increase both cell growth and protein synthesis in HeLa cells.

## APOBEC3A has modest predicted target sites in the pre-LSU rRNA

Although our overexpression assays suggest that APOBEC3A's catalytic activity is not required for its role in cell growth and protein synthesis, we wanted to more rigorously test this by analyzing the pre-rRNA for potential APOBEC3A C-to-U edit sites. The pre-rRNA/rRNA is by

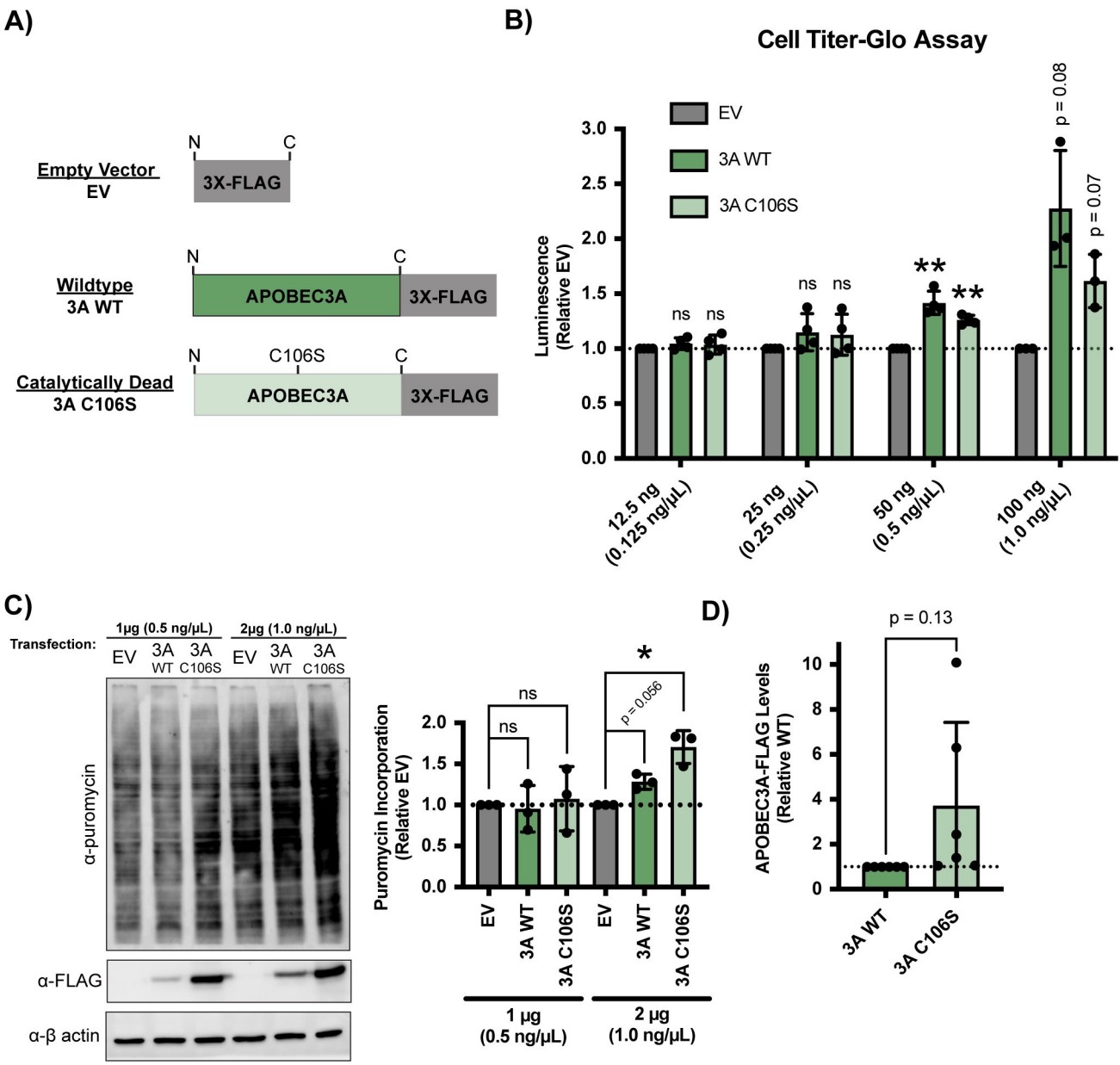

**Fig 5. Transient overexpression of APOBEC3A wild-type and catalytically dead mutated C106S increases cell growth and protein synthesis.** (A) Transfection constructs used for APOBEC3A overexpression experiments. Plasmids containing CMV promoter controlled 3X-FLAG EV top, C-terminal 3X-FLAG tagged APOBEC3A wild-type (3A WT) middle, or catalytically dead APOBEC3A C106S mutated form (3A C106S) bottom. (B) 3A WT and 3A C106S mutated form overexpression increases cell growth in HeLa cells. Quantification of Cell Titer-Glo 2.0 assay 48 h post overexpression plasmid transfection. Three or 4 biological replicates plotted mean ± SD. Data were analyzed by one-way ANOVA with Dunnett's multiple comparisons test, ** $p \leq 0.01$. (C) Higher protein levels of expressed APOBEC3A WT and C106S increase global protein synthesis in HeLa cells. After 48 h overexpression plasmid transfection, 1 μM puromycin was added for 1 h to measure translation. (Left) Representative western blot using an α-puromycin antibody. EV 1 μg and 2 μg are negative controls. α-FLAG is shown as a validation of APOBEC3A-FLAG overexpression. α-β-actin is shown as a loading control. (Right) Quantification of puromycin signal normalized to β-actin signal and relative to EV negative control. Three biological replicates plotted mean ± SD. Data were analyzed by one-way ANOVA with Dunnett's multiple comparisons test, * $p \leq 0.05$. (D) 3A C106S mutated form has a trending increase in protein levels compared to 3A WT. Quantification of α-FLAG signal normalized to β-actin signal and relative to 3A WT from (C). Six biological replicates plotted mean ± SD. Data were analyzed by Student's $t$ test, $p$-value indicated on graph. All underlying numerical values for figure found in S2 Data. EV, empty vector; SD, standard deviation.

far the most abundant RNA in the cell and APOBEC3A has already established roles in editing much lower abundance nucleic acids in cells [36–38,43–46]. APOBEC3A has a UC dinucleotide RNA sequence motif preference, where the edited cytidine is preceded by a uracil [86]. We scanned the primary 47S pre-rRNA transcript (NR_145144.1) and 5S rRNA (E00204 [87]) consensus sequences for UC sequence motifs to identify candidate portions of the pre-rRNA that APOBEC3A could bind and/or modify. Based on examining UC sequence motif occurrence at 100 nucleotide intervals across the transcript and its enrichment over its expected occurrence at random (approximately 6.25 per 100 nucleotides), we observed that this motif was more likely to occur within the ITS sequences and less likely to occur within mature rRNA sequences on the 47S pre-rRNA compared to what is expected for a random sequence (S8 Fig). More specifically, there were sites of UC motif enrichment just 5′ of cut sites 1 in the 5′ETS, cut site 2a in ITS1, and cut site 4 in ITS2, and an overall decrease below what would be predicted across the 18S rRNA and at some locations on the 28S rRNA. There were 7 UC sequence motifs identified on the 5S rRNA (S9 Fig). We kept these locations of UC motif enrichment in mind during our search for APOBEC3A target sites along the pre-rRNA, especially considering their proximity to important cleavage sites that produce the pre-LSU rRNA.

To identify predicted APOBEC3A target sites on the pre-rRNA, we performed a nuclear RNA-seq experiment followed by variant analysis comparing 3 samples [siNT and siAPOBEC3A (pool and siRNA #1)] of treated MCF10A cells in duplicate. After siRNA treatment, we harvested nuclei, extracted RNA, and performed a library prep without rRNA depletion and using random hexamer RT primers to enrich for pre-rRNA reads (S10A Fig). We obtained on average about 56 million reads and after alignment to both a 47S pre-rRNA (NR_145144.1) and 5S rRNA (E00204 [87]) consensus sequences, over half of our reads aligned to these transcripts of interest, indicating successful pre-rRNA enrichment (S10B Fig) [NCBI BioProject (Accession Number): PRJNA935922]. Upon closer inspection, our read alignments were enriched for the mature rRNAs, but still had notable coverage across both the external and internal transcribed spacer sequences (S11A and S11B Fig). This coverage was consistent across siRNA treatments within replicates (S12 Fig).

Using these datasets, we performed variant analysis using LoFreq [88] to identify C-to-U predicted target sites on the pre-rRNA that decreased after APOBEC3A depletion. This is similar to what was done previously to assess rDNA variation across individuals and tissues [89]. We identified 9 sites on the 47S pre-rRNA (none on the 5S rRNA), where a C-to-U variant was detected in both replicates in siNT treated samples with varying frequencies (0.34% to 23.39%). Out of the 9 identified edit sites, 5 had a decreased edit frequency in all the APOBEC3A depleted samples (2 replicates each of siAPOBEC3A pool and siAPOBEC3A #1), with 3 of those occurring in a UC sequence motif context (S13A Fig, Data E in S1 Data). Interestingly, all 5 of these sites occurred downstream of the pre-18S rRNA in locations corresponding to pre-LSU rRNAs or in the mature 28S rRNA. It is important to note that most of these edits were detected at a modest frequency and accompanied by a correspondingly modest, but reproducible, reduction in editing frequency upon APOBEC3A depletion (S13B Fig, Data E in S1 Data). Overall, this RNA-seq experiment has revealed a handful of predicted APOBEC3A target sites on the 47S pre-rRNA that are sensitive to the presence of the APOBEC3A, and that APOBEC3A may bind and edit.

## APOBEC3A has predicted target sites on nuclear pre-mRNAs that encode nucleolar and cell cycle regulator proteins

Although over half of the nuclear RNA-seq reads aligned to the pre-rRNA, there was still a large portion of other nuclear RNA species that APOBEC3A could regulate. This time we

aligned our reads to the hg38 human reference and noticed a prominent increase in the amount of reads that aligned to intronic sequences compared to a previously total cell RNA-seq experiment in MCF10A cells completed in our laboratory (GEO accession GSE154764) [6] (S14A and S14B Fig). Differential expression analysis comparing both siAPOBEC3A pool and siAPOBEC3A #1 to siNT treatment further confirmed our results that validated high-confidence TP53 gene targets [90] are greatly enriched among up-regulated transcripts and, within those, *CDKN1A* mRNA levels are increased (S15A and S15B Fig). As anticipated, transcripts involved in the cell cycle and cell growth, proliferation, and development pathway categories were among the most reduced after APOBEC3A depletion (S15C and S15D Fig). Next, using the same LoFreq [88] variant analysis, we identified transcriptome-wide C-to-U variants (C-to-U on + strand or G-to-A on—strand) in siNT and APOBEC3A depleted samples. As expected, there was a modest overall decrease in C-to-U variants detected in the APOBEC3A depleted samples compared to siNT (S16A Fig).

We took the top predicted APOBEC3A target sites identified for further analysis. A total of 264 sites were chosen that were detected in all 3 samples, in both replicates, and had a decreased editing frequency in both siAPOBEC3A depletion conditions (S16A Fig). Consistent with the read alignment, these top predicted target sites were detected across all portions of transcripts, but were enriched within intronic regions (S16B Fig). As noted, one key factor regulating APOBEC3A's enzymatic function is the presence of a UC sequence motif. Therefore, it was surprising that the 264 sites did not have an enrichment in preceding U presence before the edited C. However, when only looking at the top predicted targets that had a percent difference in editing frequency between siNT and siAPOBEC3A treatments of greater than 66% (Data F in S1 Data), a preceding U was the most prevalent nucleotide; however, in this same dataset, this was not a significant enrichment compared to what would be expected at random (Fisher's Exact test, $p = 0.7$) (S16C Fig). While unexpected, it is important to note that these are the first APOBEC3A candidate sites identified after nuclear enrichment, to our knowledge, due to our unique experimental design.

Examining the APOBEC3A predicted target sites in pre-mRNAs, we investigated if there were any trends in the types of transcripts they occur in. We took advantage of 3 nucleolar proteome datasets [91–94] to test if there was an enrichment of transcripts encoding nucleolar proteins over the expected at random 17.7% within our dataset. Of the 264 sites identified, they occur in 213 transcripts, 184 of which are protein coding (Data G in S1 Data). Interestingly, there is almost a 2-fold enrichment in these transcripts encoding nucleolar proteins and this increase is maintained in transcripts that contain sites that have a >33% edit change, >66% edit change, and occur within the UC sequence motif context (S17A Fig). Finally, we utilized the STRING database [95] to perform gene ontology overrepresentation analysis of the transcripts that contained our top 264 APOBEC3A candidate sites. The top overrepresented biological process categories include RNA splicing and metabolism (GO:0043484, GO:0016071, GO:0016070) and positive regulation of cell migration, negative regulation of apoptosis and cell death (GO:0030335, GO:0043066, GO:0060548) (S17B Fig). While these categories were not enriched for established nucleolar regulators, RNA regulation and apoptosis are overlapping and tightly linked to the ribosome biogenesis pathway [1,17–19]. APOBEC3A potentially targets several pre-mRNA transcripts encoding proteins present in the nucleolus and with roles regulating RNA and the cell cycle.

The most salient pre-mRNA encoding a nucleolar protein that contains an APOBEC3A candidate edit site is *DDX17* which encodes an RNA helicase required for ITS2 pre-rRNA cleavage [96]. Specifically, DDX17 contains an APOBEC3A candidate edit site within its 3′ UTR that decreased on average a modest 4.3% after APOBEC3A depletion (S18A Fig). We hypothesized that APOBEC3A's potential editing of this transcript could have impacts on

splicing, mRNA stability, and/or translation efficiency. To test this, we used siRNAs to deplete APOBEC3A and measured *DDX17* mRNA levels by qRT-PCR and protein levels of both of its alternative translation start site variants (p72 and p82) by western blotting. After APOBEC3A depletion, *DDX17* mRNA levels decreased approximately 30% but it was not statistically significant (S18B Fig). Additionally, the ratio between p72 and p82 DDX17 isoforms did not change, and we only observed a slight increase, not the suspected decrease, in both isoforms after APOBC3A depletion (S18C Fig). Based on our results, we cannot definitively state if APOBEC3A is required for the normal expression of DDX17.

## Discussion

Following up on candidate hits from a high-throughput phenotypic screen for nucleolar function [5], we defined a role for the human cytidine deaminase, APOBEC3A, in ribosome biogenesis, suggesting for the first time the intriguing possibility that nucleolar steps in making ribosomes are regulated by APOBEC3A in human cells. APOBEC3A is a strong candidate ribosome biogenesis factor as shown by validation of our previous genome-wide siRNA screen [5] for changes in nucleolar number in MCF10A cells. Biochemical assays reveal that APOBEC3A depletion leads to defects in cell cycle progression, likely through the nucleolar stress response, and decreased protein synthesis, likely due to inhibition of pre-LSU rRNA processing in the nucleus/nucleolus. We determined, through expression of a tagged APOBEC3A-FLAG in HeLa cells, that a subset of APOBEC3A is nucleolar and associates with critical ribosome biogenesis factors. Furthermore, overexpression of either wild-type or a catalytically dead version of APOBEC3A-FLAG increases cell growth and translation levels. Finally, we used a novel nuclear RNA-seq experiment indicated select putative APOBEC3A target sites in the pre-LSU rRNA and in pre-mRNAs encoding nucleolar proteins and RNA splicing and cell cycle regulators. These results strongly suggest APOBEC3A functions within the nucleus to modulate ribosome biogenesis and cell cycle progression, 2 interconnected processes, likely independent of its editing activity.

APOBEC3A-FLAG's presence within the nucleolus by IF and its association with essential nucleolar ribosome biogenesis factors FBL and RPA194 by co-IP are evidence for its direct involvement in ribosome production. While APOBEC3A's association with RPA194 might be surprising since APOBEC3A siRNA depletion did not alter pre-rRNA transcription in our studies, it is important to note that RPA194 has other interactors outside the typical transcription machinery in the nucleolus. For example, RPA194 associates with large ribosomal proteins RPL10A and RPL9 [97] and we and others have previously showed RPA194 interacts with large subunit biogenesis factors WDR12 of the PeBoW complex and RSL24D1 [75,98]. Similar to APOBEC3A, both FBL and RPA194 are protooncogenic (reviewed in [12,99,100]), highlighting their necessary role in ribosome biogenesis and cell growth. Still, the entire scope and breadth of APOBEC3A's nucleolar interactome remains to be revealed.

Furthermore, there is good correlation between APOBEC3A's role in pre-LSU rRNA processing and its predicted target sites. We observed reduced ITS2 cleavage in the pre-rRNA leading to the formation of the mature LSU 28S and 5.8S rRNAs by northern blotting. Similarly, 4 out of the 5 identified APOBEC3A C-to-U predicted target sites on the pre-rRNA are within the pre-LSU (5.8S + ITS2 + 28S rRNA). Thus, the pre-rRNA processing steps affected by siAPOBEC3A depletion correlate with the putative APOBEC3A target sites in the pre-rRNA. This, and the result that a subset of APOBEC3A localizes to the nucleolus and associates with RPA194 and FBL, points towards a direct function of APOBEC3A in pre-LSU biogenesis.

Overexpression of both the APOBEC3A WT and a catalytically dead mutated APOBEC3A C106S revealed that the editing activity of APOBEC3A is not required to drive increased cell growth and protein synthesis. This suggests a non-editing role for APOBEC3A in ribosome biogenesis. This is consistent with the fact that we only observed a low frequency of C-to-U edits in the pre-rRNA and other nuclear RNAs with only small reductions in editing frequency after APOBEC3A depletion. Very low editing frequencies and decreases in these editing frequencies when APOBEC3A is depleted may indicate that in most cases APOBEC3A is associating with these RNAs but not performing deamination. APOBEC3A's catalytic activity has been previously shown to not be required for other functions including repression of HIV-1 reactivation [82] and regulation of interferon-stimulated response elements [85]. In these examples, APOBEC3A binds ssDNA at TTTC tandem motifs to recruit epigenetic silencing machinery and suppress transcription. Likewise, our results suggest APOBEC3A's catalytic activity is not required for its role in making ribosomes.

However, enrichment of C-to-U predicted target sites on pre-mRNAs that encode proteins with nucleolar functions indicates that APOBEC3A binding or editing to the pre-mRNA could also be its indirect function in ribosome biogenesis. These APOBEC3A predicted targets in the nucleus were enriched for pre-mRNAs encoding proteins localized to the nucleolus and for proteins involved in RNA splicing (GO:0043484). Most strikingly within this subgroup of both nucleolar and RNA splicing was the pre-LSU factor *DDX17*; however, APOBEC3A depletion did not significantly alter its mRNA or protein levels. Thus, it remains to be seen the extent to which APOBEC3A regulates ribosome biogenesis indirectly by binding or editing pre-mRNAs encoding nucleolar proteins.

Is APOBEC3A targeting the pre-rRNA or the rDNA to promote ribosome production? Previous studies have observed human rDNA/rRNA variants across rDNA repeats, mature rRNAs, tissues, or individuals [89,101,102], our study did allow for identification of putative pre-rRNA variants precisely produced by 1 enzyme, APOBEC3A. While our results have not definitively answered whether our predicted targets are in RNA or DNA, our observed pre-rRNA processing and maturation defects with no changes in pre-rRNA transcription would point towards APOBEC3A functioning at the RNA level. We hypothesize that it is APOBEC3A's RNA binding ability alone that regulates pre-LSU maturation, which would explain the low edit frequencies and how the catalytically dead mutated APOBEC3A C106S promotes cell growth and protein synthesis to the same extent as WT. Lastly, increasing the likelihood that our reproducible results are from APOBEC3A's function on RNA, in cultured cancer cell lines, APOBEC3A associated genomic mutation rates are variable and episodic in nature over lengthier time periods than our experiments [40,52]. Our study serves as a first analysis of APOBEC3A in ribosome biogenesis; it is our hope that future studies will reveal more mechanistic detail regarding APOBEC3A's function within the nucleolus.

APOBEC3A's requirement in ribosome biogenesis strengthens its connection to cancer pathogenesis beyond its role in producing mutations across the genome. Previous work has established several examples of proteins working in both DNA repair and ribosome production (reviewed in [16,103]). Our work on APOBEC3A highlights another example that links the integrity of the genome to ribosome biogenesis. Currently, cytidine deaminase inhibitors are being developed (reviewed in [26]), including against APOBEC3A [104,105], with a focus on their application in cancer therapy [106]. Although, if APOBEC3A's editing activity is not required for its role in ribosome biogenesis, as our results suggest, these inhibitors would be unlikely to alter this aspect of APOBEC3A's cancer promoting ability. Even so, APOBEC3A's newfound function in ribosome biogenesis emphasizes it as a top candidate cytidine deaminase target for cancer therapeutics.

## Materials and methods

### GTEx and TCGA expression and survival dataset analysis

GTEx unmatched normal and TCGA matched normal and tumor expression datasets were obtained through the Xena platform (https://xena.ucsc.edu/) [51]. RNA-seq by Expectation-Maximization (RSEM) Log2 fold expression levels for respective cytidine deaminase transcripts were subtracted from the mean of the overall normal and tumor tissues combined and graphed. Survival data, also within the same dataset, was obtained through the Xena platform. Samples were stratified by either high or low cytidine deaminase expression and survival curves were generated. For cytidine deaminases where the median expression was 0, data was stratified by either detected or undetected levels of expression.

### Cell culture

MCF10A cells (ATCC, CRL-10317) were subcultured in Dulbecco's modified Eagles' medium/nutrient mixture F-12 (Gibco, 1130–032) containing horse serum (Gibco, 16050), 10 μg/ml insulin (Sigma, I1882), 0.5 μg/ml hydrocortisone (Sigma H0135), 100 ng/ml cholera toxin (Sigma, C8052), and 20 ng/ml epidermal growth factor (Peprotech, AF-100-15). HeLa cells (ATCC, CCL-2), cells were grown in DMEM (Gibco, 41965–062) with 10% fetal bovine serum (FBS, Gibco, 10438026). Cell lines were maintained at 37°C, in a humidified atmosphere with 5% $CO_2$.

For the high-throughput nucleolar number and 5-EU assays, 3,000 cells/well were reverse transfected in 384-well plates on day 0. For RNA or protein isolation, 100,000 cells/well were seeded into 6-well plates on day 0. For the overexpression protein isolation, 200,000 cells/well were seeded into 6-well plates on day 0. For the dual-luciferase assay, 100,000 cells/well were seeded into 12-well plates on day 0. For nuclei isolation, 1,560,000 cells were seeded into 15 cm plates on day 0. For immunofluorescence imaging, 77,460 cells/well were seeded into 4-well Nunc Lab-Tek Chamber Slide System (Thermo Fisher Scientific, 177399) on day 0. For co-immunoprecipitations, 2,440,000 cells were seeded into 10 cm plates on day 0. For the overexpression cell growth assay, 6,700 cells/well were seeded into opaque white 96-well plates (Corning Costar, 3917) on day 0. Transfections were performed on day 1 for all low-throughput experiments.

### RNAi

Horizon Discovery Biosciences siGENOME SMARTpool siRNAs and ON-TARGETplus pool and individual were used for all assays as indicated in Data B in S1 Data. siRNA transfections were completed using Lipofectamine RNAiMAX Transfection Reagent (Invitrogen, 13778150) per manufacturer's instructions with a final siRNA concentration of 20 nM for 384-well plate high-throughput assays (nucleolar number and 5-EU) and 33 nM for all other assays. For the high-throughput assays, cells were reverse transfected on day 0 and siRNA controls were added to 16 wells and other siRNAs were added to 1 well for each replicate. For all other assays cells were transfected 24 h after plating. All siRNA depletions were performed for 72 h total.

### Molecular cloning and overexpression plasmid transfection

ORF expression plasmid for APOBEC3A (NM_001270406.1, isoform b) (EX-I1284-M14) C-terminal 3X-FLAG tag was acquired from GeneCopoeia. Since this isoform contained an alternative in-frame splice site resulting in a shorter sequence, site-directed mutagenesis was performed to insert the missing exon sequence and obtain APOBEC3A (NM_145699.4, isoform a). Two other site-directed mutagenesis PCRs were performed to obtain C106S catalytically

dead mutated version of APOBEC3A and 3X-FLAG empty-vector plasmid. Cloning PCR primers are listed in supporting materials (Data H in S1 Data). Mutagenesis PCR products were self-ligated using KLD enzyme mix (New England Biolabs, M0554). Plasmid sequences were verified by Sanger sequencing (GENEWIZ/Azenta Life Sciences).

Plasmids were transfected using Lipofectamine 3000 (Thermo Fisher Scientific, L3000015) per manufacturer's instructions based on volume of media used for each experiment. Amounts of plasmid transfected are indicated except for immunofluorescence experiment (0.5 μg transfected in 500 μl, 1.0 ng/μl for 24 h) and co-immunoprecipitation experiments (12.5 μg transfected in 11.25 ml, 1.11 ng/μl for 24 h).

## Nucleolar number assay

We counted nucleolar number in MCF10A cells in high-throughput as done previously in our laboratory [5,6,75], except that siNOL11 was used as a positive control. In short, after 72 h siRNA depletion, cells were fixed, stained for nucleoli (72B9 α-fibrillarin antibody [107], AF647 goat anti-mouse IGG secondary) and nuclei (Hoechst). Images were acquired with an IN Cell 2200 imaging system (GE Healthcare) and a custom CellProfiler pipeline was used for analysis. For both siRNA pool and individual siRNA deconvolution treatments, a hit was called based on a cutoff of a mean one-nucleolus percent effect greater than or equal to +3 standard deviations (SDs) above the non-targeting siRNA (siNT) negative control (Data C and D in S1 Data) of a given replicate.

## 5-EU incorporation (Nucleolar rRNA biogenesis) assay

We measured nucleolar 5-EU incorporation (rRNA biogenesis) in high-throughput as done previously in our laboratory [71,75,108]. In short, the nucleolar number assay was followed with the addition of a 1 mM 5-ethynl uridine (5-EU) treatment at the end of a 72-h siRNA depletion. An additional click chemistry step was performed to attach an AF488 azide for 5-EU visualization. A custom CellProfiler pipeline was also used for analysis to measure median nucleolar 5-EU signal within each nucleolus.

## Cell cycle analysis

We performed cell cycle analysis as done previously [65], including in work performed in our laboratory [6]. In short, integrated Hoechst staining intensity was measured per nucleus from the high-throughput images for the 5-EU incorporation assay. The Log2 integrated intensities of each nucleus was calculated. The siNT treatment G1 peak was set at 1.0 and G2 peak at 2.0 for normalization. Cell cycle phases were defined as the following normalized Log2 integrated Hoechst staining intensities: sub-G1 < 0.75, G1 = 0.75–1.25, S = 1.25–1.75, G2/M = 1.75–2.5, >4n > 2.5.

## Immunofluorescence and colocalization

APOBEC3A-3XFLAG transfected HeLa cells were washed 2× in PBS and fixed in 1% paraformaldehyde in PBS for 20 min at room temperature. Cells were permeabilized with 0.5% Triton X-100 in PBS for 5 min at room temperature. Cells were blocked with 10% FBS in PBST (PBS with 0.1% Tween-20) for 1 h at room temperature, incubated with the following primary antibodies: α-FBL (rabbit, abcam, ab5821) and α-FLAG M2 (mouse, Thomas Scientific, C975V51) overnight at 4°C, and incubated with the following secondary antibodies: α-rabbit AF488 (Thermo Fisher Scientific, A-11034) and α-mouse AF594 (Thermo Fisher Scientific, A-11005) 1 h at room temperature. Cover glass was added over ProLong Gold

Antifade mountant with DAPI (Thermo Fisher Scientific, P36941) and cured overnight at room temperature. Images were acquired using a Zeiss LSM 880 airyscan confocal microscope. Images were analyzed and Pearson correlation coefficients were quantified on a per cell basis using FIJI ImageJ.

## Protein harvesting and western blotting

Cells were harvested by scraping followed by a PBS rinse. Cells were resuspended in AZ lysis buffer (50 mM Tris (pH 7.5), 250 mM NaCl, 1% Igepal, 0.1% SDS, 5 mM EDTA (pH 8.0)) supplemented with protease inhibitors (cOmplete Protease Inhibitor Cocktail, Roche, 11697498001) and lysed by vortexing at 4°C for 15 min. Cell debris was pelleted by spinning at 21000 RCF at 4°C for 15 min and protein containing supernatant was taken. Protein concentration was determined by Bradford assay (Bio-Rad, 5000006). An equal amount of total protein (20 to 50 µg per sample) was separated by SDS-PAGE and transferred to a PVDF membrane (Bio-Rad, 1620177) for blotting.

The following primary antibodies were used: α-APOBEC3A/B (Invitrogen, PA5-104035), α-p21-HRP (Santa Cruz, sc-6246), α-p53-HRP (Santa Cruz, sc-126), α-puromycin (Kerafast, EQ0001), α-FLAG M2-HRP (Sigma Aldrich, A8592), α-RPA194 (Santa Cruz, sc-48385), α-FBL (Abcam, ab226178), α-KAP1 (abcam, ab22553), α-PES1 (Bethyl Laboratories, A300-902A), α-LAS1L (Bethyl Laboratories, A304-438A), α-DDX17 (Santa Cruz, sc-271112), and α-β-actin (Sigma Aldrich, A1978). For detection of non-HRP conjugated primary antibodies, either secondary α-mouse (GE Healthcare, Life Sciences NA931) or α-rabbit (GE Healthcare, Life Sciences NA934V) HRP conjugated antibodies were used. Images were acquired using Bio-Rad Chemidoc (Bio-Rad, 12003153) and analyzed using ImageJ software.

## Puromycin incorporation (global protein synthesis) assay

Global protein synthesis was accessed as done previously [69], including in our laboratory [5,6,15,75]. In short, 1 µM (or 0.5 µM for Mock 0.5 control) puromycin was added for 1 h to label nascent polypeptides over that time period at the end of a 72-h siRNA depletion. Protein harvesting and western blotting was carried out as described above.

## Co-immunoprecipitation

Co-immunoprecipitation experiments were performed as done previously in our laboratory [15,75]. Briefly, protein A agarose beads (Cell Signaling Technologies, 9863) or protein A/G agarose beads (Thermo Fisher Scientific, 20421) were incubated with 20 µg of α-RPA194 (Santa Cruz, sc-48385) overnight at 4°C for conjugation. For reciprocal co-immunoprecipitations, preconjugated α-FLAG M2 agarose (Sigma Aldrich, A2220) or unconjugated Pierce control agarose (Thermo Fisher Scientific, 26150) alone were used. HeLa total cell extract was obtained by sonication and incubated with either antibody bound or unconjugated beads for 2 h at 4°C. Immunocomplexes were eluted and western blotting was performed as described above.

## RNA harvesting

RNA (either from total cell or nuclei purified fractions) was extracted using TRIzol reagent (Life Technologies, 559018) per manufacturer's instructions. RNA was washed and stored in 75% ethanol as a pellet at −20°C or −80°C before use in downstream analysis.

## Digital droplet PCR

The $A_{260/230}$ of all total cell RNA samples dissolved in nuclease-free water were determined to be above 1.7 by Nanodrop (Thermo Fisher Scientific, ND2000CLAPTOP) before proceeding to cDNA synthesis; 5 μg of each RNA sample was treated with DNase I to digest any genomic DNA. cDNA was synthesized using iScript Advanced cDNA synthesis kit (Bio-Rad, 1725037) per manufacturer's instructions. ddPCR Supermix for Probes (No dUTP) (Bio-Rad, 1863023) was used per manufacturer's instructions with gene-specific primers/probes (Data H in S1 Data). Droplets were produced using Automated Droplet Generator (Bio-Rad, 1864101). The following amplification parameters were used for PCR: 95°C for 10 min for enzyme activation, 40 cycles of 94°C for 30 s, and 55°C for 1 min, then 98°C for 10 min for enzyme deactivation. Data were acquired using QX200 Droplet Reader (Bio-Rad, 1864003). Number of positive droplets was used as a readout of mRNA expression.

## qRT-PCR

qRT-PCR was performed as carried out previously in our laboratory [75]. The $A_{260/230}$ of all RNA samples dissolved in nuclease-free water were determined to be above 1.7. cDNA was synthesized from 1 μg of total RNA using iScript gDNA clear cDNA synthesis kit (Bio-Rad, 1725035) with random primers. iTaq Universal SYBR Green Supermix (Bio-Rad, 1725121) was used to perform qPCR with gene-specific primers (Data H in S1 Data). The following amplification parameters were used: initial denaturation 95°C for 30 s, 40 cycles of 95°C for 15 s, and 60°C for 30 s. Subsequent melt curve analysis was performed to ensure a single product, 95°C for 15 s, then gradual (0.3°C/15 s) increase from 60°C to 94.8°C. Amplification of the 7SL transcript was used as an internal control and relative RNA levels were determined by using comparative $C_T$ method ($\Delta\Delta C_T$). Three technical replicates were completed for each biological replicate and averaged.

## Dual-luciferase reporter assay

rDNA promoter activity was accessed as done previously [74], including in our laboratory [5,6,75]. In short, 48 h after siRNA transfection, MCF10A cells were co-transfected with both 1,000 ng of pHrD-IRES-Luc plasmid [74] and 0.1 ng of a constitutively expressed *Renilla* containing internal transfection control plasmid [55] using Lipofectamine 3000 (Thermo Fisher Scientific, L3000015) per manufacturer's instructions. Twenty-four hours later (72 h after siRNA transfection), cells were harvested by scraping and luminescence was measured by a Dual-luciferase Reporter Assay System (Promega, E1910) per manufacturer's instructions using a GloMax 20/20 luminometer (Promega).

## BioAnalyzer RNA quantification

One μg of either total cell or nuclear fraction RNA was submitted at a concentration of 100 ng/μl in nuclease-free water to the Yale Center for Genome Analysis for Agilent BioAnalyzer analysis. Nuclear RNA samples were also used for downstream sequencing analysis (see below).

## Northern blotting

Northern blotting was performed as carried out previously in our laboratory [5,6,75]. In short, 3 μg of total cellular RNA was resolved on a denaturing 1% agarose/1.25% formaldehyde gel using Tri/Tri buffer [109]. Separated RNA was transferred to a Hybond-XL membrane (GE Healthcare, RPN303S) and UV-crosslinked. Membranes were stained with methylene blue (0.025% w/v) and imaged using Bio-Rad Chemidoc (Bio-Rad, 12003153). Blots were

hybridized to a $^{32}$P radiolabeled DNA oligonucleotide probe (ITS2, 5′–AAGGGGTCTTTAA ACCTCCGCGCCGGAACGCGCTAGGTAC– 3′) and detected using a phosphorimager (Amersham Typhoon, 29187194). Images were analyzed using ImageJ software and RAMP [76] was used for quantification.

### Nuclei harvesting

All steps were performed at 4˚C. Cells were harvested by scraping following a PBS rinse. Washed cell pellets were resuspended in Buffer A (10 mM HEPES (pH 8.0), 10 mM KCl, 1.5 mM MgCl$_2$, 0.5 mM dithiothreitol, 1X protease inhibitors (cOmplete Protease Inhibitor Cocktail, Roche, 11697498001), 4 mM NEM) and swelled for 10 min. Swollen cells were dounced using a 7 ml dounce (Wheaton, 3575420) for 20 strokes. Dounced cells were centrifuged at 220 RCF for 5 min. The pellet containing nuclei was then used for downstream RNA analysis.

### Consensus motif logo visualization

Consensus sequence motif logos were generated using WebLogo (Version 2.8.2) [112]. Input sequences included the genomic DNA sequence of 10 nucleotides upstream and downstream of candidate edited cytidines of nuclear RNAs.

### Nuclear RNA-seq and variant analysis

Library preparation and RNA-seq was completed by the Yale Center for Genome Analysis. Illumina RNA-seq library prep was completed without rRNA depletion or polyA enrichment step(s). The cDNA library was made by random primed first strand synthesis. Paired-end ($2 \times 150$ bp) reads were collected using the NovaSeq 6000 S4 XP system (Illumina).

RNA-seq analysis was completed using Partek Flow software available through the Harvey Cushing/John Hay Whitney Medical Library. For pre-rRNA analysis raw nuclear RNA-seq reads were aligned to either a 47S pre-rRNA consensus sequence (NR_145144.1) or 5S rRNA consensus sequence (E00204, 5SrRNADB [87]) as reference indexes using BowTie2 (Version 2.2.5), with default parameters. For transcriptome-wide analysis, raw nuclear RNA-seq reads were trimmed of Illumina Universal adapter sequences. Trimmed reads were aligned to the hg38 reference index using STAR (Version 2.7.8a), using default parameters.

Differential expression analysis was conducted with DESeq2 [110]. Single-nucleotide variants (SNVs) were detected using LoFreq [88] (Version 2.1.3a) against the respective aligned reference above, default parameters, $p < 0.05$, read depth >10. Data are available on NCBI: accession PRJNA935922.

Variant analyses (location, edit type, amino acid consequence) was performed using Ensembl Variant Effect Predictor [111].

Comparisons were made to a previously obtained total MCF10A cell RNA-seq dataset completed in our laboratory (GEO accession GSE154764) [6].

### Enrichment analyses

A curated high confidence list of 343 TP53 gene targets was obtained from [90]. siAPOBEC3A up-regulated nuclear transcripts were cross referenced to this list and compared to a baseline of the 343 high-confidence genes that have been found in human genome (43,768 HGNC approved genes [113], 0.8%).

Qiagen Ingenuity Pathway Analysis was used to analyze enriched pathways and upstream regulators present in the nuclear transcripts that were up- and down-regulated after

siAPOBEC3A depletion. Ingenuity knowledge base (genes only) was used as a reference dataset with default settings.

A metadataset of nucleolar proteins was created based on 3 previous proteomic nucleolar datasets [91–94]. Adding all the proteins together that were detected in at least one of these datasets results in 3,490 nucleolar proteins out of 19,670 total in the entire human proteome (17.7%) [94], which was used as a baseline for enrichment analysis.

STRING Database (Version 11.5) [95] was used for biological processes gene ontology enrichment analysis. Categories were reported where fold enrichment $> 2$ and $p < 0.05$.

### Statistical analyses

Statistical analyses were performed using GraphPad Prism 9.3.1 (GraphPad Software). Tests are described in the associated figure legends.

## Supporting information

**S1 Fig. Nine out of the 11 human cytidine deaminases are more highly expressed in tumor versus normal tissue.** Violin plots from Genotype-Tissue Expression (GTEx) unmatched normal and The Cancer Genome Atlas (TCGA) matched normal and tumor RNA-seq by Expectation-Maximization (RSEM) [51] Log$_2$ fold expression levels for human cytidine deaminases subtracted from the mean. Mean of tumor ($N = 9,185$) and normal ($N = 7,429$) cytidine deaminase expression set at 0 (horizontal line). For each indicated transcript, median (heavy dashed line), quartiles (light dashed line), normal data (light shading), and tumor data (darker shading). Percent change in expression between normal versus tumor indicated for each cytidine deaminase transcript. Data were analyzed by Student's $t$ test, significant $p$-values are reported on the graph. All underlying numerical values for figure found in S2 Data.
(TIF)

**S2 Fig. Seven out of the 11 human cytidine deaminases are more highly expressed in tumor versus normal tissue and their higher expression is correlated with a decrease in survival.** (A) Higher expression of 8 out of the 11 human cytidine deaminases correlates with decreased survival probability. Survival data was obtained from Genotype-Tissue Expression (GTEx) unmatched normal samples and The Cancer Genome Atlas (TCGA) matched normal and tumor samples using the Xena platform [51]. Samples were stratified by either high or low cytidine deaminase expression and survival curves were generated. For cytidine deaminases where the median expression was 0, data was stratified by either detected or undetected levels of expression. Median survival (days) and the percent difference between low and high expression groups were reported. Data were analyzed by Kaplan–Meier survival analysis, significant $p$-values reported on respective graphs. (B) Seven out of the 11 human cytidine deaminases exhibit higher expression in tumor versus normal tissue and higher expression is associated with lower survival probability. Percent change in mRNA expression from (S1) (x-axis) and percent change in median survival from (A) (y-axis) was plotted for each cytidine deaminase. An increase in expression and a decrease in survival quadrant are indicated by the gray background. All underlying numerical values for figure found in S2 Data.
(TIF)

**S3 Fig. Deconvoluted individual siRNAs targeting APOBEC3A reduce protein levels and global protein synthesis in MCF10A cells.** (A) Validation of APOBEC3A protein depletion after siAPOBEC3A individual siRNA #1 and #2 treatment. (Left) Representative western blot using an α-APOBEC3A/B antibody. siNT is a negative control. α-β-actin is a loading control. (Right) Quantification of APOBEC3A protein levels normalized to β-actin signal and relative

to siNT negative control. Three biological replicates plotted mean ± SD. Data were analyzed by Student's *t* test, ** $p \leq 0.01$, * $p \leq 0.05$. (B) Depletion with individual siAPOBEC3A #1 and #2 reduces global protein synthesis in MCF10A cells. After 72 h siRNA depletion, 1 μM puromycin was added for 1 h to measure translation. (Left) Representative western blot using an α-puromycin antibody. Mock and siNT are negative controls, siRPL4 is a positive control, and Mock 0.5 μM is a control to indicate robust quantification. α-β-actin is shown as a loading control. (Right) Quantification of puromycin signal normalized to β-actin signal and relative to siNT negative control. Four biological replicates were plotted mean ± SD. Data were analyzed by one-way ANOVA with Dunnett's multiple comparisons test, * $p \leq 0.05$. All underlying numerical values for figure found in S2 Data.
(TIF)

**S4 Fig. siAPOBEC3A pool treatment reduces protein levels and global protein synthesis in HeLa cells.** (A) Validation of APOBEC3A protein depletion after siAPOBEC3A pool treatment in HeLa cells. (Left) Representative western blot using an α-APOBEC3A/B antibody. siNT is a negative control. α-β-actin is shown as a loading control. (Right) Quantification of APOBEC3A protein levels normalized to β-actin signal and relative to siNT negative control. Three biological replicates plotted mean ± SD. Data were analyzed by Student's *t* test, ** $p \leq 0.01$. (B) siAPOBEC3A pool treatment reduces global protein synthesis in HeLa cells. After 72 h siRNA depletion, 1 μM puromycin was added for 1 h to measure translation. (Left) Representative western blot using an α-puromycin antibody. Mock and siNT are negative controls, siRPL4 is a positive control, and Mock 0.5 μM is a control to indicate robust quantification. α-β-actin is shown as a loading control. (Right) Quantification of puromycin signal normalized to β-actin signal and relative to siNT negative control. Three biological replicates plotted mean ± SD. Data were analyzed by one-way ANOVA with Dunnett's multiple comparisons test, ** $p \leq 0.01$, * $p \leq 0.05$. All underlying numerical values for figure found in S2 Data.
(TIF)

**S5 Fig. APOBEC3A is required for nucleolar rRNA biogenesis but not pre-rRNA transcription.** (A, B) siAPOBEC3A depletion (pool) modestly reduces nucleolar rRNA biogenesis in MCF10A cells. (A) After 72 h siRNA depletion, 1 mM 5-EU was added for 1 h to measure nucleolar rRNA biogenesis. Representative images of nucleoli stained with α-fibrillarin (FBL, red), 5-ethynl uridine (5-EU) visualized by click-chemistry attached AF488 azide (green), nuclei stained with DAPI (blue), and FBL and 5-EU merged (yellow). siNT is a negative control and siPOLR1A (large subunit of RNA Polymerase 1) is a positive control. *N* = number of nuclei (cells) analyzed in all 3 biological replicates and median nucleolar 5-EU signal (nucleolar rRNA biogenesis) reported. (B) Quantification of nucleolar rRNA biogenesis percent inhibition. siNT negative control is set to 0% and siPOLR1A positive control is set to 100%. Based on results in Bryant and colleagues [71], factors required for pre-rRNA transcription have a percent inhibition > ~80% (red background) and factors only required for pre-rRNA processing/maturation have a percent inhibition approximately 50% to 80% (gray background). Three biological replicates are plotted mean ± SD. (C) Schematic of 47S+45S pre-rRNA transcript measured by qRT-PCR in (D) using the indicated primers (orange). (D) siAPOBEC3A depletion (pool) does not significantly reduce 47S+45S pre-rRNA transcript levels. qRT-PCR was performed to measure primary 47+45S pre-rRNA transcript levels. Mock and siNT are negative controls and siPOLR1A is a positive control; $2^{-\Delta\Delta CT}$ measured relative to 7SL internal control and siNT negative control. Three technical replicates of 3 biological replicates plotted mean ± SD. Data were analyzed by Student's *t* test, * $p \leq 0.05$. (E) Schematic of luciferase reporter plasmids to measure rDNA promoter activity [74]. (Top) Firefly pHrD-IRES-Luc rDNA promoter reporter plasmid. (Bottom) *Renilla* luciferase constitutive promoter reporter

plasmid, transfection control. (F) siAPOBEC3A depletion (pool) does not significantly reduce rDNA promoter activity in MCF10A cells. After 48 h of siRNA depletion, plasmids in (E) were transfected for 24 h for a total of 72 h siAPOBEC3A depletion. Firefly luminescence was measured relative to *Renilla* luminescence and the siNT negative control. siPOLR1A (the large subunit of RNA Polymerase 1) is a positive control. Three biological replicates plotted mean ± SD. Data were analyzed by Student's *t* test, ** $p \leq 0.01$. All underlying numerical values for figure found in S2 Data.

(TIF)

**S6 Fig. siAPOBEC3A pool treatment does not change LAS1L levels in MCF10A cells.** (Left) Representative western blot using α-LAS1L antibody. siNT is a negative control. α-β-actin is shown as a loading control. (Right) Quantification of LAS1L protein levels normalized to β-actin signal and relative to siNT negative control. Three biological replicates plotted mean ± SD. Data were analyzed by Student's *t* test. All underlying numerical values for figure found in S2 Data.

(TIF)

**S7 Fig. HeLa cells are more easily transfected with CMV promoter controlled APO-BEC3A-3XFLAG plasmid.** (A) Representative western blot using α-FLAG antibody on HeLa whole cell lysate. Increasing amounts of APOBEC3A-3XFLAG plasmid transfected for 24 h in 10 cm dishes (10 ml media volume). Stain-free total protein is shown as a loading control. One biological replicate. (B) Same as in (A) except using MCF10A whole cell lysate. (C) Same as in (B) except including higher amounts of plasmid.

(TIF)

**S8 Fig. APOBEC3A's UC dinucleotide sequence motif is more frequently found in the external and internal transcribed spacer (ETS and ITS) sequences near pre-rRNA cleavage sites and less frequently found in the 18S rRNA coding sequence.** The occurrence of UC sequence motifs was calculated for every 100 nucleotide (nt) interval across the pre-rRNA and graphed. The occurrence of UC sequence motif at random over 100 nt is predicted to be 6.25, indicated by the dotted horizontal line. Pre-rRNA cleavage sites are indicated with vertical solid lines. All underlying numerical values for figure found in S2 Data.

(TIF)

**S9 Fig. The human 5S rRNA contains 7 UC dinucleotide sequence motifs.** Secondary structure of the human 5S rRNA (E00204) obtained from 5SrRNAdb [87]. UC sequence motifs highlighted in green. Expected number of UC sequence motifs reported based on occurrence at random of 6.25 per 100 nucleotides.

(TIF)

**S10 Fig. Schematic of the nuclear RNA-seq experimental design and alignment results.** (A) Schematic of nuclear RNA-seq experiment. MCF10A cells were treated with siNT negative control, siAPOBEC3A pool, or siAPOBEC3A individual #1 for 72 h. Nuclear RNA was extracted and submitted for sequencing with no rRNA depletion step and primed with random hexamers for reverse transcription (RT). (B) Over half of the nuclear RNA-seq reads aligned to the 47S pre-rRNA and to the 5S rRNA. Average reads, percent alignment, and coverage depth of all nuclear RNA-sequencing runs using Bowtie 2 alignment to either the 47S pre-rRNA (NR_145144.1) or 5S rRNA (E00204, [87]). Average of 6 biological replicates (2x siNT, 2x siAPOBEC3A pool, 2x siAPOBEC3A #1) ± SD.

(TIF)

 

**S11 Fig. Nuclear RNA-sequencing reads spanned the entire 47S pre-rRNA with enrichment in the mature rRNA sequences compared to the external and internal transcribed spacer pre-rRNA sequences.** (A) Coverage report across the 47S pre-rRNA sequence (hg38, chromosome 21) after STAR (Version 2.7.8a) alignment to hg38. Y-axis is average read depth; x-axis is genomic coordinates on chromosome 21 corresponding to the location of the 47S pre-rRNA (RNA45SN1). Data is for nuclear RNA-seq of siNT negative control treated MCF10A cells replicate #1. Image was generated using Partek Flow chromosome viewer. (B) Same as in (A) except zoomed into the different portions of the 47S pre-rRNA transcript including the 5′ and 3′ external transcribed spacers (ETSs), internal transcribed spacers (ITSs) 1 and 2, and the mature (18S, 5.8S, and 28S) rRNAs. Cleavage sites indicated below the x-axis. (TIF)

**S12 Fig. Nuclear RNA-sequencing read coverage of the 47S pre-rRNA and 5S rRNA was consistent across siRNA treatments within replicates.** (A) Coverage summary of MCF10A cell nuclear RNA-seq reads aligned to a 47S pre-rRNA consensus sequence (NR_145144.1) for all replicates indicated by color. Y-axis is average coverage depth; x-axis is the entire 47S pre-rRNA sequence. Image was generated using Partek Flow chromosome viewer. (B) Same as in (A) except for the 5S rRNA (E00204 [87]) sequence. (TIF)

**S13 Fig. APOBEC3A is enriched for predicted target sites on the pre-LSU rRNA.** (A) LoFreq variant analysis of the nuclear RNA-seq datasets identified 5 predicted C-to-U APOBEC3A target sites in the pre-rRNA (green down arrows). Three of these occur at a UC sequence motif (green highlighted squares). Table of 9 C-to-U edit locations on the pre-rRNA detected by LoFreq (see Methods) in siNT negative control. Editing frequencies are the mean ± SD of 2 biological replicates. A change in the edit frequency after siAPOBEC3A depletion was called if there was a decrease in editing after siAPOBEC3A pool and siAPOBEC3A #1 depletion in both replicates for each (down green arrow). If not, it was indicated as a non-APOBEC3A target site (horizontal gray arrows). (B) siAPOBEC3A depletion leads to modest but reproducible decreases in editing frequency at 5 predicted C-to-U variant sites on the pre-rRNA. Percent differences between siAPOBEC3A edit frequency and siNT negative control edit frequency (dotted horizontal line) at C-to-U predicted target sites in (A) are graphed. Two biological replicates of siAPOBEC3A pool (circle data points) and siAPOBEC3A #1 (triangle data points) are plotted as the mean (horizontal line) ± SD. Locations on the pre-rRNA in nucleotides are indicated on x-axis. (C) Schematic of 47S pre-rRNA, 5 green stars indicate locations of predicted APOBEC3A C to U target sites from (A, B). All underlying numerical values for figure found in S2 Data. (TIF)

**S14 Fig. MCF10A nuclear RNA-seq reads aligned to mostly intronic regions of pre-mRNAs.** (A) Average reads, percent alignment, percent coverage percentage, and coverage depth of all nuclear RNA-sequencing runs using STAR alignment to the hg38 reference index. Average of 6 biological replicates (2x siNT negative control, 2x siAPOBEC3A pool, 2x siAPOBEC3A #1) ± SD. (B) Nuclear RNA-seq reads largely align to introns of pre-mRNAs. Exon and intron distribution of MCF10A nuclear RNA-seq read alignments (this experiment) vs. total RNA-seq (rRNA depleted) read alignment (GEO accession GSE154764) [6]. Percentage of total for each location type is shown in a pie chart and in a table based on color. (TIF)

**S15 Fig. Differential expression analysis of nuclear transcripts after siAPOBEC3A depletion reveals induction of TP53 targets and decreases in cell cycle and cell growth regulator transcripts.** (A) siAPOBEC3A pool and siAPOBEC3A #1 treatment induced nuclear

 

transcripts are enriched for TP53 gene target transcripts by RNA-seq, and 343 genes were determined to be high confidence targets of TP53 through analysis of several datasets [90] out of 43,768 HGNC approved human genes [113], 0.8%. siAPOBEC3A pool treatment up-regulated 29 TP53 target nuclear transcripts (>2-fold up-regulated, $p \leq 0.05$, FDR ≤0.05) out of 278 total up-regulated transcripts. siAPOBEC3A #1 treatment up-regulated 41 TP53 target transcripts (>2-fold up-regulated, $p \leq 0.05$, FDR ≤0.05) out of 323 total up-regulated nuclear transcripts. Percent of TP53 targets present in genome, siAPOBEC3A pool up-regulated nuclear transcripts and siAPOBEC3A #1 up-regulated nuclear transcripts are graphed. Data were analyzed by Fisher's exact test, **** $p \leq 0.0001$. (B) siAPOBEC3A treatment increases *CDKN1A* nuclear transcript levels in MCF10A cells by RNA-seq. *CDKN1A* FPKM for 2 biological replicates of negative control siNT, siAPOBEC3A pool, and siAPOBEC3A #1. (C) Qiagen Ingenuity Pathway Analysis of siAPOBEC3A pool differentially expressed nuclear transcripts, and 443 down-regulated transcripts (>2-fold, $p \leq 0.05$, FDR ≤0.05, blue) and 278 up-regulated transcripts (>2-fold, $p \leq 0.05$, FDR ≤0.05, orange). (Left) Chart of enriched pathways where -log($p$-value) > 3.0. Top 7 pathways are labeled. Size of circle represents ratio of overlapping genes represented in dataset. Up-regulated pathways are in orange and down-regulated pathways are in blue where darkness of shade indicates the extent to which pathway is regulated. (Right) Table of top 15 positive upstream regulators enriched in up-regulated transcript list. TP53 and CDKN1A are bolded in green. Molecule type and (positive) Z-score shown. (D) Same as in (C) except for siAPOBEC3A #1 treatment, and 440 down-regulated transcripts (>2-fold, $p \leq 0.05$, FDR ≤0.05, blue) and 323 up-regulated transcripts (>2-fold, $p \leq 0.05$, FDR ≤0.05, orange). All underlying numerical values for figure found in S2 Data. (TIF)

**S16 Fig. siAPOBEC3A depletion leads to reduced predicted C-to-U edits, primarily within intronic regions.** (A) There are 264 predicted APOBEC3A C-to-U edit sites that were revealed by LoFreq variant analysis of nuclear RNAs. The total number of all single-nucleotide variants (SNVs) reported by LoFreq was reported as an average of 2 biological replicates ± SD. Total and percent of total C-to-U variants were reported as an average of 2 biological replicates ± SD (percent difference from siNT negative control). C-to-U variants were filtered for those that were detected in both replicates of all 3 samples (i.e., all 6 biological replicates) and those that decreased on average for both siAPOBEC3A pool and siAPOBEC3A #1 depletions. (B) APOBEC3A predicted target sites on nuclear RNAs are found mostly within intronic regions. The distribution of the 264 predicted APOBEC3A target sites in (A) location types are reported as a percentage of total in a pie chart. Locations which include 5′ or 3′ untranslated regions (UTRs), transcript types (coding or non-coding), and edit type in parentheses (intron, missense, non-sense, synonymous, non-coding exon) are indicated by color. (C) The APOBEC3A UC sequence motif is not significantly enriched in the top predicted target sites. (Top) sequence logos for all 264 predicted target sites in (A) including 10 nucleotides upstream and downstream of the edited C. (Bottom) Same as (Top) except indicating only the top 16 predicted target sites (exhibiting a greater than 66% decrease in siNT editing frequency compared to the average of the siAPOBEC3A pool and siAPOBEC3A #1 treatments) were used for logo creation. (TIF)

**S17 Fig. APOBEC3A has predicted target sites on pre-mRNA transcripts that encode nucleolar proteins and cell cycle regulators.** (A) The APOBEC3A predicted target sites are enriched on pre-mRNAs encoding nucleolar proteins. The 264 predicted target sites were located on 213 transcripts of which 184 are protein coding. An estimate of human nucleolar proteins were determined based on the presence of a protein in at least 1 of 3 proteomic datasets ($N = 3,490$) [91–94]. The total number of human proteins was reported based on Thul and

colleagues ($N$ = 19,670) [94]. Proteins that were encoded by transcripts with APOBEC3A predicted targets and subgroups within were tested for enrichment of nucleolar protein coding transcripts by comparison to an estimate of percentage of the proteome that is nucleolar (17.7%, dotted horizontal line) and graphed. (B) The APOBEC3A predicted target sites are enriched on pre-mRNAs involved in RNA metabolism, positive regulators of cell migration, and negative regulators of cell apoptosis/death. All 213 transcripts that contain APOBEC3A predicted target sites were analyzed for overrepresentation of biological function gene ontology categories using the STRING database [95]. Categories were reported where fold-enrichment > 2.0 and $p < 0.05$, and $p$-values indicated on graph. All underlying numerical values for figure found in S2 Data.
(TIF)

**S18 Fig. DDX17 contains an APOBEC3A predicted target site but its levels do not significantly change after APOBEC3A depletion on the mRNA or protein level.** (A) DDX17 (NM_006386) contains an APOBEC3A predicted target site within its 3′ UTR. (B) siAPOBEC3A depletion (pool) does not significantly reduce *DDX17* mRNA levels. qRT-PCR was performed to measure primary DDX17 mRNA levels. Mock and siNT are negative controls; $2^{-\Delta\Delta CT}$ measured relative to 7SL internal control and siNT negative control. Three technical replicates of 3 biological replicates plotted mean ± SD. Data were analyzed by Student's $t$ test, $p$-value indicated on graph. (C) siAPOBEC3A depletion (pool) does not reduce DDX17 protein levels or change its isoform ratio. (Left) Representative western blot using an α-DDX17 antibody. α-β-actin is shown as a loading control. (Right) Quantification of DDX17 (p72 and p82 isoforms) protein levels normalized to β-actin signal and relative to siNT and quantification of p82/p72 isoform ratio. Three biological replicates plotted mean ± SD. Data were analyzed by Student's $t$ test, ** $p \leq 0.01$. All underlying numerical values for figure found in S2 Data.
(TIF)

**S1 Data. (separate file).** (A) Previously published human ribosome biogenesis related screening data for cytidine deaminases. Data includes ranking within respective screen, if it was a published hit (based on reported cutoff), or is a potential hit (based on an arbitrary cutoff) in the process that was tested. (B) List of siRNAs used in this study. (C) siON-TARGET one-nucleolus rescreening data for APOBEC3A and APOBEC4. (D) siON-TARGET one-nucleolus siRNA deconvolution screening data for APOBEC3A and APOBEC4. (E) 47S pre-rRNA (NR_145144.1) variants identified by LoFreq. (F) Top 16 nuclear transcriptome-wide (hg38) variants identified by LoFreq that are APOBEC3A predicted target sites. (G) All nuclear transcriptome-wide (hg38) variants identified by LoFreq that are APOBEC3A predicted target sites. (H) Primers and probe sequences used in this study.
(XLSX)

**S2 Data. (separate file).** Numerical values for all plotted data (Figs 1, 2, 3, 4, 5, S1, S2, S3, S4, S5, S6, S8, S13, S15, S17 and S18).
(XLSX)

**S1 Raw Images. (separate file).** Original, uncropped, and minimally adjusted blot and gel images.
(PDF)

## Acknowledgments

We thank the Yale Center for Genome Analysis (YCGA) for performing Agilent Bioanalyzer analysis, RNA-seq library prep, and sequencing. We thank Peter M. Glazer's laboratory for

their help with digital droplet PCR protocols and use of their equipment, specifically Kelly E. W. Carufe. We thank Reuben S. Harris for helpful discussion and insights regarding APOBEC3A's known biological functions and experiment ideas. We thank Matthew D. Simon and Isaac W. Vock for the helpful discussion and insight regarding RNA sequencing protocols and data analysis. We thank Rolando Garcia-Milian of the Harvey Cushing/John Hay Whitney Medical Library Bioinformatics Support Hub for helpful teaching about using Partek Flow software and RNA-seq data visualization. We thank members of the Baserga laboratory for their helpful discussions and insight during the preparation and writing of this manuscript as well. We acknowledge the use of CellProfiler for image analysis (http://www.cellprofiler.org/).

## Author Contributions

**Conceptualization:** Mason A. McCool, Susan J. Baserga.

**Data curation:** Mason A. McCool, Carson J. Bryant, Laura Abriola, Susan J. Baserga.

**Formal analysis:** Mason A. McCool, Carson J. Bryant, Susan J. Baserga.

**Funding acquisition:** Mason A. McCool, Susan J. Baserga.

**Investigation:** Mason A. McCool, Carson J. Bryant, Susan J. Baserga.

**Methodology:** Mason A. McCool, Carson J. Bryant, Susan J. Baserga.

**Project administration:** Laura Abriola, Yulia V. Surovtseva.

**Resources:** Laura Abriola, Yulia V. Surovtseva, Susan J. Baserga.

**Supervision:** Yulia V. Surovtseva, Susan J. Baserga.

**Validation:** Mason A. McCool, Susan J. Baserga.

**Visualization:** Mason A. McCool.

**Writing – original draft:** Mason A. McCool.

**Writing – review & editing:** Mason A. McCool, Carson J. Bryant, Susan J. Baserga.

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
