## [Editor Report · Decision Letter 0]

16 Mar 2023

Dear Dr Baserga, 

Thank you for submitting your manuscript entitled "The cytidine deaminase APOBEC3A is required for large ribosomal subunit biogenesis" for consideration as a Research Article by PLOS Biology.

Your manuscript has now been evaluated by the PLOS Biology editorial staff, as well as by an academic editor with relevant expertise, and I am writing to let you know that we would like to send your submission out for external peer review.

Once your full submission is complete, your paper will undergo a series of checks in preparation for peer review. After your manuscript has passed the checks it will be sent out for review. To provide the metadata for your submission, please Login to Editorial Manager (https://www.editorialmanager.com/pbiology) within two working days, i.e. by Mar 18 2023 11:59PM.

Kind regards,

Richard

Richard Hodge, PhD

Associate Editor, PLOS Biology

rhodge@plos.org

PLOS

---

## [Decision Letter · Decision Letter 1]

25 Apr 2023

Dear Dr Baserga,

Thank you for your patience while your manuscript "The cytidine deaminase APOBEC3A is required for large ribosomal subunit biogenesis" was peer-reviewed at PLOS Biology. Please accept my apologies for the delays that you have experienced during the peer review process. Your manuscript has been evaluated by the PLOS Biology editors, an Academic Editor with relevant expertise, and by three independent reviewers.

The reviews are attached below. As you will see, the reviewers find your study interesting and well done, but raise overlapping concerns with the strength of the data directly linking APOBEC3A to ribosome biogenesis, as well as the lack of deeper mechanistic data that provides insights into how APOBEC3A regulates ribosome assembly. The reviewers suggest several experiments to address these concerns, including the addition of rescue assays with catalytically inactive mutants and investigating whether APOBEC3A interacts with ribosome assembly factors or pre-rRNA and its role in editing mRNA/rRNA.

Based on their specific comments and following discussion with the Academic Editor, it is clear that a substantial amount of work would be required to meet the criteria for publication in PLOS Biology. However, given our and the reviewer interest in your study, we would be open to inviting a comprehensive revision of the study that thoroughly addresses all the reviewers' comments. Given the extent of revision that would be needed, we cannot make a decision about publication until we have seen the revised manuscript and your response to the reviewers' comments. Your revised manuscript would need to be seen by the reviewers again, but please note that we would not engage them unless their main concerns have been addressed. 

We appreciate that these requests represent a great deal of extra work, and we are willing to relax our standard revision time to allow you 6 months to revise your study. Please email us (plosbiology@plos.org) if you have any questions or concerns, or envision needing a (short) extension.

**IMPORTANT**

After discussions with the academic editor and the rest of the editorial team, we also think that your study would be a good fit as a Discovery Report (https://journals.plos.org/plosbiology/s/what-we-publish). ‘Discovery Reports’ describe novel and intriguing initial findings with the potential to lead to a significant new result for the field. Discovery Reports are short articles, typically with 2-4 main figures. In this format, we feel that the revision would need to provide the rescue experiments to show a direct link between APOBEC3A and ribosome biogenesis to strengthen the conclusions and some preliminary interaction assays with ribosome assembly factors or prerRNA. 

Please do not hesitate to contact me about the preferred route you would like to go down for the revision and if you have any further questions or concerns about this. 

**IMPORTANT - SUBMITTING YOUR REVISION**

*Resubmission Checklist*

*Published Peer Review*

*PLOS Data Policy*

*Blot and Gel Data Policy*

Sincerely,

Richard

Richard Hodge, PhD

Associate Editor, PLOS Biology

rhodge@plos.org

REVIEWS:

Reviewer #1: This manuscript follows up on previous work from the Baserga Lab identifying novel regulators of ribosome biogenesis (Farley-Barnes, Cell Rep 2018). In a previous high-throughput siRNA screen the authors identified APOBEC3A and APOBEC4 as hits based on the observation that their depletion leads to a one-nucleolus effect. Because APOBEC3A had stronger and more consistent results with the siRNA screen the authors chose to investigate its role in ribosome biogenesis further. The authors found that knockdown of APOBEC3A leads to changes in cell cycle progression, induction of the nucleolar stress response, and a decrease in global protein synthesis. Next the authors investigated if APOBEC3A plays a role in pre-rRNA transcription and ribosome assembly. Results from both a 5-EU incorporation assay and pre-rRNA transcription reporter assay were consistent with APOCEC3A playing a role in ribosome assembly downstream of POLI transcription. Quantification of RNA and northern blots from knockdown cells revealed that loss of APOBEC3A leads to 60S specific effects on pre-rRNA processing including a buildup of the 32S pre-rRNA and a decrease in the mature 28S rRNA. 

The authors hypothesized that APOBEC3A could regulate ribosome assembly through either its C-to-U editing activity or its RNA binding activity. To explore this further the authors identified putative editing sites within the pre-rRNA and then performed RNA-seq experiments, which revealed putative editing sites within the pre-rRNA. The authors also identified putative editing sites in mRNAs encoding for nucleolar and cell cycle regulation genes. In the first four figures of the manuscript the authors present very convincing evidence that APOBEC3A is a regulator of the maturation of the large ribosomal subunit. However, they provide no concrete evidence for how APOBEC3A does this and if it is a direct or in-direct effect. More experimental evidence would be needed for the authors to propose that APOBEC3A edits pre-rRNA and/or nuclear pre-mRNA transcripts. 

1. The authors could begin to tackle the direct vs indirect question through rescue experiments with catalytically inactive APOBEC3A. 

2. Given the strong association between APOBEC3A and cancer do the authors think that APOBEC3A's role in regulating ribosome assembly is specific to cancer cells (only MCF10A and HeLa cell lines were tested)?

3. Apobec3A has established RNA binding activity. Is there any evidence to suggest it can associate with nascent pre-60S particles or other known regulators of ribosome assembly?

4. The sequencing results presented in Figure 5 are confusing. I do not understand the graph presented in 5E. Are the significant editing sites those above the dashed line? Moreover, the edit frequencies observed are very low, making it difficult to understand how this could have a direct impact on ribosome assembly. 

5. Without direct evidence of C-to-U editing the authors should tone down some of their claims such as the following sentence (line 524) "Revealing for the first time the intriguing possibility that the pre-rRNA is edited …"

Reviewer #2: Ribosome assembly is a complex pathway that is upregulated in cancer cells to support the increased demand for translation. However, ribosome assembly is energy intensive and the pathway is normally strictly regulated according to cellular needs and nutrient availability. A deep understanding of the ribosome assembly pathway in humans may identify potential targets for drug therapies against cancers. Previously, the Baserga lab screened for factors that altered nucleolar morphology to identify novel ribosome assembly factors. Among their previous hits were APOBEC3A and APOBEC4, cytidine deaminases previously not known to be involved in ribosome assembly. Here, the authors more closely examine the potential roles of these deaminases in ribosome assembly. The experimental data are clearly presented and the experiments appear to be well-executed. The authors provide preliminary results suggesting APOBEC3A impacts assembly of the large subunit. However, their characterization of how APOBEC3A affects ribosome assembly, whether through editing the rRNA, partaking directly in the assembly pathway or editing the mRNA of a ribosomal protein gene or assembly factor or editing another RNA species such as snoRNAs, is lacking. Overall, the analysis is quite preliminary and does not provide new mechanistic insights into ribosome assembly. Considerable additional experimental evidence is needed to substantiate the authors' primary claim, that APOBEC3A plays a crucial role in 28S assembly and to provide insight into its mechanism of action in ribosome assembly. Additionally, the writing needs careful editing for logical connection of ideas.

Major points:

1) The authors characterize the phenotype of cells after knockdown of APOBEC3A. The authors should show that an si-resistant APOBEC3A can rescue the knockdown. In addition, the authors should determine if a catalytic mutant of APOBEC3A can rescue to determine if it is the editing function of APOBEC3A that is required, or if the protein plays a chaperoning role as seen for many RNA modifying enzymes in ribosome assembly.

2) The two primary pieces of evidence that APOBEC3A has a role in ribosome assembly are: a) knockdown of APOBEC3A leads to elevated levels of 32S pre-rRNA intermediate and b) to reduced levels of 28S. This is a good starting point for additional analysis. For example, the authors should determine if APOBEC3A interacts with ribosome assembly factors. This could be done by co-IP or Bio-ID and mass spec. If such factors or interactions can be identified, then the authors should ask if these interactions are important for ribosome assembly.

3) Fig 2. In panel C, the authors show that knockdown of APOBEC3A leads to reduced puromycin incorporation, a surrogate for translation. However, a similar result would result from any of numerous cell stress conditions leading to accumulation of monosomes. The authors should show polysome profiles of these cells to establish that these cells contain polysomes.

4) Fig 3 suggests that APOBEC3A is not involved in transcription. This figure is more appropriate for supplemental material.

5) Fig 5 deals with the identification of edits in rRNA. The authors conclude in panel A that the spacer regions are enriched for edit sites. However, another interpretation appears to be that 18S and 28S rRNAs experience less editing than surrounding sequence. This should be addressed.

6) Although the authors have identified apparent edits in rRNA, their frequency is exceptionally low. For those in 28S that appear to be APOBEC3A-dependent, at most, only 1.1% of the RNA is edited. This cannot account for the substantial loss of 28S. The accumulation of 32S implies a defect in cleavage within ITS2 and the only APOBEC3A-dependent site in ITS2 is edited at less than 0.5%, again, not explaining the loss of 28S. Furthermore, the change in editing frequency upon APOBEC3A knockdown is very modest, less than 10%. Collectively, these results do not establish that rRNA editing by APOBEC3A impacts ribosome assembly. Rather, they suggest that rRNA editing is not important for assembly.

7) In Figure 6, the authors turn to the possibility that APOBEC3A editing of pre-mRNA or mRNA can explain its role in ribosome assembly. Here, the authors identify a multitude of candidate genes and call out DDX17 as an intriguing candidate because it has a known role in ribosome assembly. However, the authors do not do any experiments to test the function of APOBEC3A editing this transcript or any others in their candidate list. Again, this is a good starting point for additional directed analysis which needs to be done.

8) In various places, the logic presented seems confused. One example: L35-38: The authors try to connect the mutagenic role of deaminases in cancer with their potential role in ribosome assembly. This writing confuses the issues. Mutagenesis by deaminases is not driving ribosome assembly and ribosome assembly is needed for but does not drive cancer. The role that the authors propose for deaminases in ribosome assembly is unrelated to and conceptually distinct from their role in cancer.

Minor points

1) The logogram in Fig 6E should show RNA sequence (eg, U not T).

2) L243: This is not correctly stated: "Since the nucleolar stress response is caused by a reduction in mature ribosome levels…" Nucleolar stress leads to reduced levels of mature ribosomes but I am not aware of evidence that reducing the levels of mature ribosomes leads to nucleolar stress. (And this is not the conclusion of the citations.)

3) L530: It is an overstatement to write: "we defined a role for APOBEC3A in ribosome biogenesis in the nucleus…"

4) L532: The authors claim: "We discovered that the human cytidine deaminases, APOBEC3A and APOBEC4, are candidate ribosome biogenesis factors through validation of our previous genome-wide siRNA screen." This holds for APOBEC3A but not APOBEC4.

---

## [Decision Letter · Decision Letter 2]

6 Feb 2024

Dear Dr Baserga,

Thank you for your patience while we considered your revised manuscript "The cytidine deaminase APOBEC3A promotes human ribosome biogenesis" for publication as a Research Article at PLOS Biology. Please accept my apologies for the delay in getting back to you during this round of the peer review process. Your revised study has been evaluated by the PLOS Biology editors, the Academic Editor and the original reviewers.

The reviews are attached below. As you can see, whilst Reviewer #1 and #3 are now generally satisfied and positive about the revision, Reviewer #2 continues to raise concerns with the overall strength of the evidence supporting a non-catalytic role for APOBEC3A during ribosome biogenesis. Whilst we appreciate that overexpression experiments using a non-catalytic APOBEC3A mutant were included in the revised version, the reviewer notes that these assays do not directly address the question of whether APOBEC3A impacts ribosome biogenesis via its editing activity. In addition, the reviewers raise overlapping concerns with the framing of the study around the editing activity of APOBEC3A and the imbalance in the discussion section when discussing its rRNA editing activity.

After discussing the reviews at length with the Academic Editor, given that all three reviewers requested this specific experiment in the previous round, we do agree with Reviewer #2 that the knockdown rescue with a non-catalytic mutant is necessary for the publication of the manuscript at the journal, in order to more directly demonstrate a non-editing role for APOBEC3A in ribosome assembly.

In light of the reviews, we will not be able to accept the current version of the manuscript, but we would welcome re-submission of a much-revised version that takes into account the reviewers' comments. We cannot make any decision about publication until we have seen the revised manuscript and your response to the reviewers' comments.

We expect to receive your revised manuscript within 3 months. Please do not hesitate to contact me directly (rhodge@plos.org) if you have any questions or concerns about the revision, or would like to request an extension. At this stage, your manuscript remains formally under active consideration at our journal; please notify us by email if you do not intend to submit a revision so that we may end consideration of the manuscript at PLOS Biology.

**IMPORTANT - SUBMITTING YOUR REVISION**

*Re-submission Checklist*

*Published Peer Review*

*PLOS Data Policy*

*Blot and Gel Data Policy*

Sincerely,

Richard

Richard Hodge, PhD

rhodge@plos.org

REVIEWS:

Reviewer #1: During the revisions the authors have provided several additional experiments aimed at deciphering the role of APOBEC3A in ribosome assembly. Using an overexpression system the authors demonstrate that overexpression of both WT and catalytic deficient APOBEC3A increases cell growth and protein synthesis, suggesting that the editing activity of APOBEC3A is not required for ribosome assembly. Immunofluorescence staining and co-IP analysis further establish that APOBEC3A is present in the nucleolus and associates with several nucleolar proteins. While the authors have some data to support a modest role of APOBEC3A in editing pre-rRNA, the additional experiments strongly support a non-editing role for APOBEC3A in ribosome assembly. Overall this manuscript is well written and provides compelling evidence that APOBEC3A plays an important role in promoting ribosome assembly.

Minor Comment:

In Fig. 5C it would appear that the C105S mutation expresses at a much higher rate than WT APOBEC3A, suggesting that this mutation may increase protein stability. Have similar observations been made with catalytic mutants of other APOBEC proteins?

Reviewer #2: In this revision the authors have added significant new material to address concerns raised from the previous submission. They have also revised the text to address prior issues. While the authors have clearly established that APOBEC3A is required for efficient ribosome biogenesis, a major point that I and the other two reviewers had raised regarded the question of whether or not the catalytic activity of APOBEC3a was required for this function. This question begins to get at the mechanism by which APOBEC3A acts. The straightforward way to address this question suggested by all three reviewers was to complement the knockdown of APOBEC3A with a catalytic mutant. In this revision, the authors have chosen to overexpress WT and catalytic mutant APOBEC3A and they show that overexpression of either promotes cell growth and translation. This is a rather different matter which raises other issues while not addressing the question at hand. I am puzzled by why the authors did not do the obvious experiment suggested by all three reviewers. As such, this does not adequately address my previous concerns. Many other minor issues have now been addressed in this revision.

Major points

1. The authors clearly showed that knocking down APOBEC3A resulted in reduced ribosome assembly. They showed this by several means, altered number of nucleoli, altered rRNA processing and reduced levels of mature 28S rRNA, all of which led to reduced levels of translation. An important mechanistic question was whether or not these effects were due to the catalytic activity of APOBEC3A, implying that editing an RNA target causes these changes. This would be a really interesting finding. This question could easily be addressed to a first approximation by determining if siRNA-resistant WT and catalytic defective APOBEC3A could rescue the knockdown, assayed by counting number of nucleoli and measuring 28S/18S levels by Bioanalyzer, as they did previously. Oddly, the authors chose rather to overexpress APOBEC3A. Despite the rationale they give for using overexpression, it is not necessarily expected that if knocking down APOBEC3A reduces ribosome levels then overexpressing APOBEC3A should increase levels. Ribosome assembly and its cellular control is highly complex and it is extremely unlikely that overexpression of a single biogenesis factor, especially one not intimately involved regulating rDNA transcription, will upregulate the pathway. More likely, overexpression of APOBEC3A modulates cell growth through other pathways and conclusions based on overexpression of APOBEC3A do not necessarily address the mechanistic question of whether APOBEC3A impacts ribosome biogenesis through editing a target substrate.

2. Regarding, rRNA editing, as noted in the earlier draft, the edit frequency for sites in 28S or ITS2 that were altered upon APOBEC3A depletion are all 1% or less with percent difference upon depletion 10% or less. It is difficult to imagine how these very modest changes can account for the reported impact on ribosome assembly. Considering the new conclusion that APOBEC3A's editing function is not important for its role in ribosome biogenesis, the entire section on rRNA editing now appears to be rendered moot. Perhaps this section should now be supplemental.

Minor points

3. In Fig 1A, siNOL11 shows a percent effect >100 but is described as being set to 100%. This needs explanation.

4. Table 1 should be moved to the supplemental material as there is no indication that any of these targets has a role in modulating ribosome biogenesis through APOBEC3A function.

5. Regarding editing of the DDX17 mRNA, as the authors note, a decrease of 4% is modest at best. Considering that follow up work could not show significance of this edit, the DDX17 work does not significantly contribute to the manuscript and should be deleted.

Reviewer #3: The authors have substantially revised their manuscript which is now entitled "The cytidine deaminase APOBEC3A promotes human ribosome biogenesis". The manuscript now includes several additional experiments in response to reviewers' comment. Importantly, the authors showed that overexpression of wt APOBEC3A creates the inverse phenotype to depleting it, namely an increase in growth and increased protein synthesis. Importantly, the same phenotype is observed when overexpressing catalytically inactive APOBEC3A suggesting that C-to-U editing is not required for this phenotype and likely not for the effects of APOBEC3CA on ribosome biogenesis. Moreover, the authors strengthened the evidence linking APOBEC3A to ribosome synthesis by showing that a subset of the protein is located in the nucleolus and that it interacts with certain ribosome biogenesis factors. Taken together, these data now convincingly demonstrate that APOBEC3A plays a role in ribosome biogenesis that is independent of its catalytic activity and thus its mutagenic activity. The authors have accordingly revised the text and in particular the discussion. 

Overall, the manuscript's quality is significantly improved, and I have only one major and a few minor suggestions.

Major suggestion:

1. The weakest point of the manuscript is the low level of editing detected by the authors. I appreciate that they have now removed the claim that editing is directly linked to ribosome biogenesis based on the new data with a catalytically inactive APOBEC3A variant. However, their remains an imbalance in the discussion where the authors FIRST discuss on 1.5 pages the editing activity which is misleading to the reader. Only in the remaining one page do the authors briefly discuss APOBEC3A's non-catalytic role in ribosome biogenesis which is the main, and most exciting finding of their study. I strongly recommend reversing the order in the discussion and expand on the possible mechanisms how APOBEC3A could play a role in ribosome biogenesis. For example, the authors do not comment on the significance of APOBEC3A's interactions with ribosome biogenesis factors and their functions.

Minor suggestions:

1. L278-284: the authors need to re-phrase the sentences and correct the logic. Currently, they stated that "factors only required (for?) pre-rRNA transcription or maturation result in lower percent inhibition values ~ 50-80%". This makes sense and the authors report 50.8% nucleolar rRNA biogenesis percent inhibition. However, the next sentence contradicts these statements as the authors conclude "it is likely that APOBEC3A is not required for pre-rRNA transcription".

2. L382: what is meant with "IF"? I assume the authors refer to immunoprecipitation.

3. Fig. 5C and D show the same data with the only difference being the reference point for normalization. This is misleading, and Fig 5D should be removed.

4. Fig. 7E: in the results (L598f), the authors claim that "there was a slight enrichment of a preceding U present". This statement is not justified by the logo shown in Fig. 7E, and the fact that it is based on only 16 top sequences. The authors must remove this claim.

5. L693: The sentence should clarify that "it" means "APOBEC3A" and that "addition of siRNAs" means "addition of control siRNAs (siNT)".

6. L702: the authors should clarify that targeted deep sequencing will only uncover DNA editing for mRNAs, but not for rRNA due to the large number of rDNA genes.

7. L708: when discussing cytidine deaminase inhibitors as potential future cancer therapeutics, the authors must also discuss that such inhibitors would likely not alter APOBEC3A's effect on ribosome biogenesis which is independent of the catalytic activity.

---

## [Decision Letter · Decision Letter 3]

23 May 2024

Dear Dr Baserga,

Thank you for your patience while we considered your revised manuscript "The cytidine deaminase APOBEC3A promotes human ribosome biogenesis" for publication as a Research Article at PLOS Biology. I am Suzanne, an Associate Editor covering for Richard while he is on parental leave. The revised version of your manuscript has been evaluated by the PLOS Biology editors, the Academic Editor and two of the original reviewers.

Based on the reviews, we are likely to accept this manuscript for publication, provided you satisfactorily address the following data and other policy-related requests.

*We would like to suggest a different title to improve accuracy: "The cytidine deaminase APOBEC3A promotes cell growth and ribosome biogenesis via the regulation of pre-mRNAs encoding nuclear proteins"

*DATA POLICY:

Regardless of the method selected, please ensure that you provide the individual numerical values that underlie the summary data displayed in the following figure panels as they are essential for readers to assess your analysis and to reproduce it: 1DEF, 2BC, 3ADGH, 4B, 5BCD.

*CODE POLICY

Per journal policy, if you have generated any custom code during the curse of this investigation, please make it available without restrictions upon publication. Please ensure that the code is sufficiently well documented and reusable, and that your Data Statement in the Editorial Manager submission system accurately describes where your code can be found. 

*BLOT AND GEL REPORTING REQUIREMENTS:

We require the original, uncropped and minimally adjusted images supporting all blot and gel results reported in an article's figures or Supporting Information files. We will require these files before a manuscript can be accepted so please prepare and upload them now. Please carefully read our guidelines for how to prepare and upload this data: https://journals.plos.org/plosbiology/s/figures#loc-blot-and-gel-reporting-requirements

We expect to receive your revised manuscript within two weeks. 

*Published Peer Review History*

*Press*

Sincerely,

Suzanne

Suzanne De Bruijn, PhD,

Associate Editor

sbruijn@plos.org

PLOS Biology

Reviewer remarks:

Reviewer #2: The authors have clearly tried to get the rescue experiment to work. It's unfortunate that is did not work. Nevertheless, the authors have adjusted their conclusions accordingly and I believe the results merit publication.

Reviewer #3 (Ute Kothe): Following the reviewers' feedback, the authors have further revised their manuscript entitled "The cytidine deaminase APOBEC3A promotes human ribosome biogenesis". While they added a number of important experiments in the first round of revisions, I appreciate that the authors have now substantially improved the interpretation of their data and the representation of their findings. Overall, the authors are now taking a much more cautious approach highlight both the findings and limitations of their study. As ribosome biogenesis, its regulation and the interesting link to DNA metabolism are very complex processes, this cautionary approach is appropriate as it better guides the reader in comparing the findings presented here to current and future work in related fields. 

In the response to reviewer 2, the authors clearly acknowledge that it would have bee preferable to rescue the phenotype of APOBEC3A knockdown by expressing in particular a catalytically inactive variant. It seems that they have made substantial efforts towards this goal but did not succeed. This is unfortunate, but nevertheless the authors have generated interesting insight into the link of APOBEC3A to ribosome biogenesis. Therefore, I support publication of this manuscript in this current form.

On the other hand, it raises the question why the rescue experiments failed and whether more complex biological mechanisms are at play. Ideally, such "negative" results should also be published to inform the research community, but I acknowledge that this is not (yet) custom in our field.

In conclusion, the revised manuscript appropriately and critically presents interesting new findings into the role of APOBEC3A for ribosome biogenesis. I recommend publishing this manuscript.

---

## [Editor Report · Decision Letter 4]

20 Jun 2024

Dear Dr Baserga,

Thank you for the submission of your revised Research Article "The cytidine deaminase APOBEC3A regulates nucleolar function to promote cell growth and ribosome biogenesis" for publication in PLOS Biology. On behalf of my colleagues and the Academic Editor, Jeff Coller, I am pleased to say that we can in principle accept your manuscript for publication, provided you address any remaining formatting and reporting issues. These will be detailed in an email you should receive within 2-3 business days from our colleagues in the journal operations team; no action is required from you until then. Please note that we will not be able to formally accept your manuscript and schedule it for publication until you have completed any requested changes.

PRESS

Sincerely, 

Suzanne De Bruijn, PhD, 

Associate Editor

PLOS Biology

sbruijn@plos.org